# MITIGATING DIFFUSION MODEL HALLUCINATIONS WITH DYNAMIC GUIDANCE

## ABSTRACT

Diffusion models, despite their impressive demos, often produce hallucinatory samples with structural inconsistencies that lie outside of the support of the true data distribution. Such hallucinations can be attributed to excessive smoothing between modes of the data distribution. However, semantic interpolations are often desirable and can lead to generation diversity, thus we believe a more nuanced solution is required. In this work, we introduce **Dynamic Guidance**, which tackles this issue. Dynamic Guidance *mitigates* hallucinations by selectively sharpening the score function only along the pre-determined directions known to cause artifacts, while preserving valid semantic variations. To our knowledge, this is the first approach that addresses hallucinations at generation time rather than through post-hoc filtering. Dynamic Guidance substantially reduces hallucinations on both controlled and natural image datasets, significantly outperforming baselines.

## 1 INTRODUCTION

Diffusion models have emerged as the dominant paradigm for image generation (Ho et al., 2020; Rombach et al., 2022; Chen et al., 2024) due to their ability to generate high-fidelity and diverse images. Despite their success, they still remain prone to generating hallucinations (Aithal et al., 2024); i.e., samples that could never appear in the training set and are outside the support of the theoretical data distribution. A typical example in natural images are samples with incorrect anatomy, such as human hands with the incorrect number of fingers. Hallucinations are also studied in the context of text-image misalignment (Liu et al., 2023; Qin et al., 2024; Lim et al., 2025; Li et al., 2025) (Zhao et al., 2025), but these approaches do not address hallucinations as a fundamental issue in diffusion model sampling.

Aithal et al. (2024) attributed hallucinations to mode interpolation, showing that models generate samples that lie between incompatible modes, i.e., regions of high probability density in the data distribution, producing semantically invalid content. They trace the mode interpolation issue to the learned score function of the diffusion model being excessively smooth between modes of the data distribution; the true score function is significantly sharper in the intermediate regions between modes, which means that the required denoising steps in those regions are significantly larger. They verify that mode interpolation does not occur when using the closed-form score on a mixture of Gaussians setting.

To avoid those low-probability regions, the denoising process must take larger steps, similar to the ones that would have been taken if the true score function had been used. Conventional guidance methods, such as classifier (Dhariwal & Nichol, 2021) and classifier-free (Ho & Salimans, 2021) guidance, are designed to steer samples toward high-likelihood regions of the data distribution, typically to improve sample quality in conditional generation. They essentially sharpen the score function in directions that correspond to the given condition. Hallucinations arise in low-likelihood regions, motivating the use of guidance not only for improving fidelity but also for mitigating hallucinations during sampling.

Guidance with a pre-determined, fixed condition does not account for the full sampling trajectory; when the guidance condition and the initial noise are misaligned, the sample can be pushed into regions that require large corrective steps. The error in these steps scales with their magnitude (Figures A.11, A.12), which can cause the trajectory to overshoot or undershoot the target mode (Section 5).

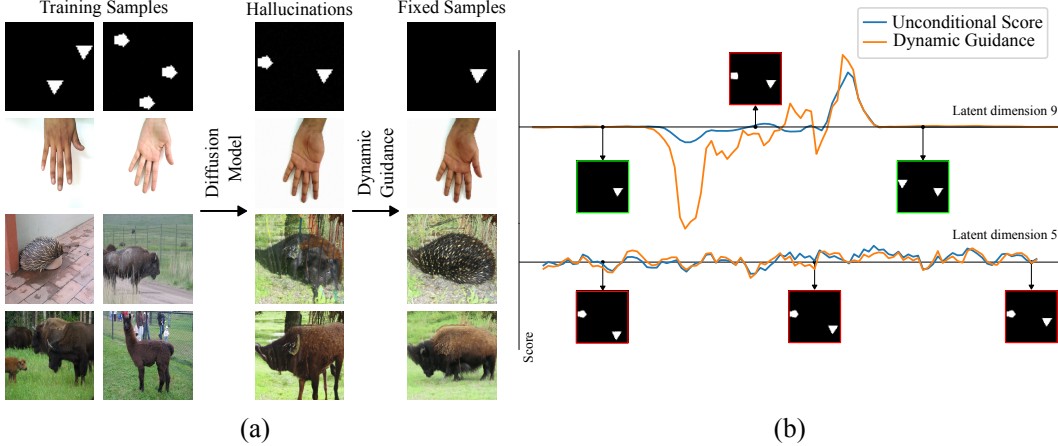

Figure 1: **(a)** Examples of training images, hallucinations, and corresponding samples fixed by Dynamic Guidance; **(b)** We pick an initial image that contains two different shapes (triangle + square), a hallucination for the Single Shapes dataset. We focus on a latent dimension that controls the appearance of the left shape **(Top)**. To resolve the hallucination, the square on the left should disappear or turn into a triangle. In the in-between region, where the left shape is square or pentagon, the unguided score function is zero, "trapping" the sample and generating a hallucination. Dynamic Guidance sharpens the score in this region, steering the sample toward valid images that only contain triangles. Dynamic Guidance does not affect the score function along dimensions that are unrelated to hallucinations, like the one controlling the position of the shape on the right **(Bottom)**.

We also argue that we should be selective in reducing interpolations between modes **along the specific directions** where hallucinations occur. Some of the interpolations are welcome: they correspond to valid semantic variation and are essential for maintaining diversity in the generated samples (i.e., model generalization (Deschenaux et al., 2024)), while others lead to implausible generations, perceived as hallucinations. For instance, in the latent manifold of hand images, interpolation along directions corresponding to skin tone yields valid and diverse samples, while interpolation along directions that control finger position may generate anatomically inconsistent hands with an incorrect number of fingers (Figure 1). We demonstrate that the score function can be sharpened *selectively* by choosing guidance classes such that interpolations between them correspond precisely to hallucinations, thereby suppressing invalid generations without reducing diversity elsewhere (Section 4.2).

We propose Dynamic Guidance, which adaptively selects the target for guidance at each denoising step. Instead of committing to a fixed condition from the start, a noisy-sample classifier is used to identify the most likely mode given the current state, and the guidance is applied toward that mode. By allowing the target to change throughout sampling, Dynamic Guidance avoids being locked into trajectories which require large, interpolation-producing steps. Dynamic Guidance is the *first method that mitigates hallucinations during the sampling process itself*, rather than relying on post-hoc detection and rejection of flawed samples. Crucially, by intervening directly in the generative process, our approach prevents hallucinations from arising in the first place. Mitigating hallucinations during sampling is preferable to post-hoc detection because it avoids wasting compute on samples that will later be discarded and preserves diversity by keeping the desired interpolations.

We evaluate Dynamic Guidance in diverse settings: **(i)** a 2D Gaussian toy dataset, where the closed form score function is available, showing that Dynamic Guidance sharpens the learned score to more accurately approximate the true score; **(ii)** two controlled image datasets of geometric shapes, where Dynamic Guidance selectively sharpens the score along latent dimensions tied to shape appearance, while leaving dimensions tied to position unaffected; **(iii)** a dataset of human hands, where hallucinations can be precisely defined and measured; and **(iv)** the large-scale ImageNet benchmark, where hallucinations cannot be precisely specified, and we therefore rely on measuring proxy metrics that are negatively impacted by hallucinations, such as generative precision (Kynkäänniemi et al., 2019) and Inception Score (Salimans et al., 2016). In all cases, Dynamic Guidance yields clear improvements over prior detection-based and static guidance methods. Dynamic Guidance is also the only

approach that significantly reduces hallucinations (by at least 50%) using the practical few-step DDIM (Song et al., 2020), rather than depending on the slow DDPM sampling.

## 2 RELATED WORK

**Diffusion Models**   Introduced by Sohl-Dickstein et al. (2015) diffusion models (Ho et al., 2020; Song et al., 2021b) are a class of generative models characterized by a forward process, where gaussian noise is gradually added in an originally clean sample, and a reverse process, where a denoising network learns to iteratively remove noise from the noisy sample, essentially learning to convert noise to data. Diffusion models have various interpretations. Score-based generative modeling (Song & Ermon, 2019; Song et al., 2021a) and DDPMs (Ho et al., 2020) are works developed concurrently that propose similar formulations of generative models that relate to score matching (Hyvärinen & Dayan, 2005; Vincent, 2011). Song et al. (2021b) proposed a framework that generalizes both using stochastic differential equations (SDEs). This framework models both the forward and backward processes as continuous-time SDEs that can be solved using a numerical SDE solver.

**Guiding Diffusion Models**   Several works have proposed methods to control the reverse diffusion process to generate samples with conditional constraints (Dhariwal & Nichol, 2021; Ho & Salimans, 2021; Graikos et al., 2022). Dhariwal & Nichol (2021) introduce an external classifier into the diffusion model's reverse process, using its gradients to influence the sampling trajectory towards a target class or attribute (classifier guidance), while (Ho & Salimans, 2021) jointly train a conditional and an unconditional diffusion model and sample from a mix of their estimated scores (classifier-free guidance). Those guidance methods aim to improve the fidelity of the generated samples by strategically sampling from well-learned high probability regions. A more recent work (Karras et al., 2024) suggests that it is possible to improve the fidelity of the generations by guiding the model with a weaker one, trained on the same data and objective.

**Hallucinations in Diffusion Models**   Prior work has examined hallucinations in diffusion models Aithal et al. (2024); Qin et al. (2024); Lim et al. (2025), but these are most often defined in terms of text–image misalignment (Li et al., 2025; Liu et al., 2023; Borji, 2023), where the diffusion model fails to generate an image that matches the conditioning prompt. Such hallucinations are more linked to issues of language–vision alignment and text conditioning, and most approaches address them by improving prompt adherence through post-training(Mrini et al., 2024; Hu et al., 2025; Le et al., 2025). In contrast, we take a view of hallucinations that is more directed to the diffusion model: rather than treating them as failures of the text condition and the diffusion model's adherence to it, we study them as a fundamental property of the diffusion model itself. Our focus is on the model's score function and its tendency to interpolate across modes in ways that yield unrealistic samples, independent of the text condition.

**Mode Interpolation**   Aithal et al. (2024) analyze the problem of hallucinations in diffusion models through mode interpolation. This, however, is not the first time the phenomenon of mode interpolation has been observed in generative models, and not all instances of mode interpolation result in hallucinations. Deschenaux et al. (2024) show that you can guide a diffusion model to produce interpolations of *desired* attributes not present in the training data, while there are other recent works that detect "mode mixture" in GANs (An et al., 2020) and diffusion models (Li et al., 2023).

## 3 BACKGROUND

**Denoising Diffusion Probabilistic Models**   DDPMs (Ho et al., 2020) learn to draw samples from a given data distribution $q(\boldsymbol{x})$. They consist of a forward process, in which Gaussian noise of increasing variance, controlled by a pre-defined schedule $\bar{\alpha}_t$, is iteratively added to a sample $\boldsymbol{x}_0 \sim q(\boldsymbol{x})$ to produce a noisy sample $\boldsymbol{x}_t$, and a reverse process that learns how to denoise samples by predicting the added noise with a neural network $\epsilon_\theta(\boldsymbol{x}_t, t)$.

Once the denoising network is trained, DDPMs can draw new samples, starting from random Gaussian noise $\boldsymbol{x}_T \sim \mathcal{N}(\boldsymbol{0}, \boldsymbol{I})$, by following the inverse process transitions

$$p_\theta(\boldsymbol{x}_{t-1}|\boldsymbol{x}_t) = \mathcal{N}(\boldsymbol{x}_{t-1}; \mu_\theta(\boldsymbol{x}_t, t), \Sigma_\theta(\boldsymbol{x}_t, t)), \tag{1}$$

where the mean is given from the predicted noise $\epsilon_\theta(\boldsymbol{x}_t, t)$ and the variance $\Sigma_\theta(\boldsymbol{x}_t, t))$ can either be fixed or learned (Nichol & Dhariwal, 2021; Dhariwal & Nichol, 2021).

**Denoising Diffusion Implicit Models**  Song et al. (2020) introduced DDIM, which allows deterministic sampling from a trained diffusion model with fewer steps. We adopt this sampling approach when using fewer than 100 sampling steps, following Nichol & Dhariwal (2021). In DDIM, the reverse sampling steps are defined as:

$$x_{t-1} = \sqrt{\bar{\alpha}_{t-1}} \underbrace{\left( \frac{x_t - \sqrt{1 - \bar{\alpha}_t}\, \epsilon_\theta(x_t, t)}{\sqrt{\bar{\alpha}_t}} \right)}_{\text{``predicted } \boldsymbol{x}_0\text{''}} + \underbrace{\sqrt{1 - \bar{\alpha}_{t-1}}\, \epsilon_\theta(x_t, t)}_{\text{``direction pointing to } x_t\text{''}}, \tag{2}$$

where the model uses a *prediction* of the clean image to perform the denoising.

**Connections to Score Based Generative Models**  The score function $s(x)$ of a probability distribution $p(x)$ is defined as the gradient of the log probability density function, $s(x) = \nabla_x \log p(x)$. Score-based generative modeling (Song & Ermon, 2019) aims to learn this score function of the data distribution from samples drawn from the same distribution. In the context of diffusion models, denoising diffusion has been shown to also approximate the score function (Song et al., 2021b)

$$s_\theta(x_t, t) = -\frac{\epsilon_\theta(\boldsymbol{x}_t, t)}{\sqrt{1 - \bar{\alpha}_t}}. \tag{3}$$

**Classifier Guidance**  Data generated by diffusion models often fail to reproduce the clarity of the training data. A widely-used technique to increase fidelity of samples is classifier guidance (Dhariwal & Nichol, 2021), which uses the gradient of a classifier $p(y|\boldsymbol{x}_t)$, trained on noisy samples $\boldsymbol{x}_t$, to guide the denoiser network towards synthesizing more realistic samples. Classifier guidance modifies the predicted noise from a network with a term that maximizes the classifier likelihood

$$\epsilon'_\theta(\boldsymbol{x}_t, t) = \epsilon_\theta(\boldsymbol{x}_t, t) - \lambda\sqrt{1 - \bar{\alpha}_t}\nabla_{\boldsymbol{x}_t} \log p(y \mid \boldsymbol{x}_t), \tag{4}$$

where $\lambda$ is the scale hyperparameter, controlling the strength of the guidance.

# 4 DYNAMIC GUIDANCE

Aithal et al. (2024) show that the learned score function in DDPMs is overly smooth in low-density regions between modes, which can "trap" samples during denoising. This issue is particularly pronounced when sampling with fewer steps; the denoiser might not be able to take large enough steps to escape, producing a hallucination between modes. On the other hand, conventional guidance methods, such as classifier (Dhariwal & Nichol, 2021) and classifier-free guidance (Ho & Salimans, 2021), are designed to steer samples toward high-likelihood regions of the data distribution, typically to improve sample quality in conditional generation. Since hallucinations arise in low-likelihood regions, this motivates the use of guidance not only for improving sample fidelity but also for mitigating hallucinations during sampling.

We propose **Dynamic Guidance**, a simple yet effective guidance algorithm that adaptively sharpens the learned score function along trajectories where hallucinations are likely to arise. The core idea is to pick class labels that correspond to the defined hallucinations and apply classifier guidance dynamically, without fixing the guidance target/condition at the beginning of the sampling process.

Depending on the dataset, we can choose class labels whose interpolations semantically align with the defined hallucinations (Section 4.1: Single Shapes). In cases where this is not possible, we can still select labels that are loosely related to hallucinations (Section 4.1: Mixed Shapes). In both settings, Dynamic Guidance is effective even when the label–hallucination relationship is not direct.

At each timestep, we identify the class with the maximum log probability given current noisy sample $\boldsymbol{x}_t$. We then perform the sampling step by applying classifier guidance using the selected class:

$$\hat{\epsilon} = \epsilon_\theta(\boldsymbol{x}_t) - \lambda\sqrt{1 - \bar{\alpha}}\nabla_{\boldsymbol{x}_t} p(y^*|\boldsymbol{x}_t), \quad y^* = \underset{y}{\arg\max} \log p(y|\boldsymbol{x}_t). \tag{5}$$

By recalculating the most probable class at each timestep, we allow the classifier guidance to remain aligned with the local score direction, adapting to the sample's evolving trajectory. The proposed

Dynamic Guidance algorithm for DDIM sampling is summarized in Algorithm 1. The extension to DDPM is straightforward and is included in the appendix Section A.1.

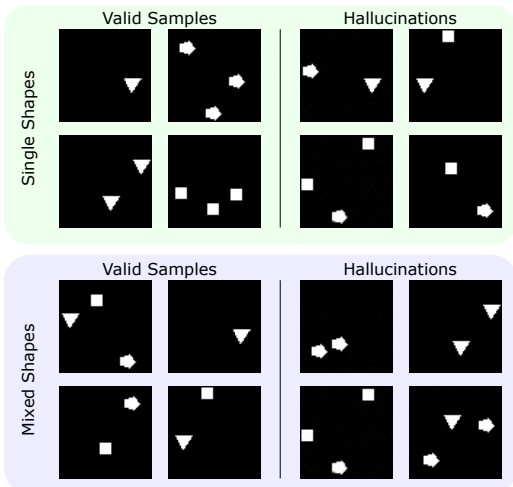

---

**Algorithm 1** Dynamic Guidance with DDIM

**Input:** timesteps $T$, dynamic guidance steps $T_1$ $T_2$, denoiser $\epsilon_\theta$, classifier $p_\phi$
$\boldsymbol{x}_T \sim \mathcal{N}(0, \boldsymbol{I})$
**for** $t = T \dots 1$ **do**
  **if** $T_1 \geq t \geq T_2$ **then**
    $y^* = \arg\max_y \log p_\phi(y|\boldsymbol{x}_t)$
    $\hat{\epsilon} = \epsilon_\theta(\boldsymbol{x}_t) - \lambda\sqrt{1-\bar{\alpha}}\nabla_{\boldsymbol{x}_t} p_\phi(y^*|\boldsymbol{x}_t)$
  **else**
    $\hat{\epsilon} = \epsilon_\theta(\boldsymbol{x}_t)$
  **end if**
  $\tilde{\boldsymbol{x}}_0 = \frac{1}{\sqrt{\bar{\alpha}_t}}\left(\boldsymbol{x}_t - \sqrt{1-\bar{\alpha}_t}\hat{\epsilon}_\theta(\boldsymbol{x}_t, t)\right)$
  $\boldsymbol{x}_{t-1} = \sqrt{\bar{\alpha}_{t-1}}\tilde{\boldsymbol{x}}_0 + \sqrt{1-\alpha_{t-1}}\,\hat{\epsilon}_\theta(\boldsymbol{x}_t, t)$
**end for**
**return** $\boldsymbol{x}_0$

---

Figure 2: Examples of valid samples and hallucinations for the Single Shapes **(Top)** and Mixed Shapes **(Bottom)** datasets.

### 4.1 HALLUCINATIONS & CLASS LABELS

In this section, we describe the datasets used for our study and define what is considered a hallucination for each one. We also point to the class labels used for our Dynamic Guidance.

**2D Mixture of Gaussians:**   We create a synthetic toy dataset with a mixture of 25 2D Gaussians arranged in a square grid, similar to Aithal et al. (2024). Since the true distribution $p(\boldsymbol{x}_0)$ is known in this setup, we define hallucinations as samples $\boldsymbol{x}$ for which $p(\boldsymbol{x}) < p(\boldsymbol{x}_{4\sigma})$, where $\boldsymbol{x}_{4\sigma}$ is a threshold of 4 standard deviations away from the nearest Gaussian component. We cluster the generated dataset into 25 clusters and use the cluster assignments as labels for Dynamic Guidance.

**Single Shapes:**   We construct a synthetic image dataset consisting of images containing triangles, squares, and pentagons. Each image contains one to three instances of the same shape, but never multiple shape types. In this setting, we define hallucinations as images containing different shapes simultaneously (see Figure 2). The class labels $c$ are given by the shape type, $c \in \{T, S, P\}$. Hallucinations, therefore, correspond to interpolations across these labels.

**Mixed Shapes:**   We adopt the synthetic image dataset from Aithal et al. (2024) consisting of images of triangles, squares, and pentagons. Each image contains up to one instance of each shape, and may contain multiple shape types. In this setting, we define hallucinations as images containing 2 or more instances of the same shape (see Figure 2). The class labels $c$ are given by the combination of shapes, $c \in \{T, S, P, TS, TP, SP, TSP\}$. Hallucinations in this case do not necessarily correspond to interpolations across the selected class labels.

**Hands:**   We use the Hands-11k dataset (Afifi, 2019), which contains images of human hands. Here, we define hallucinations as images that deviate from the expected hand anatomy. For Dynamic Guidance, we use the 4 classes provided by the dataset, corresponding to the orientation of the hand: *"palmar right"*, *"palmar left"*, *"dorsal right"*, and *"dorsal left"*. Interestingly, in this setting, some hallucinations directly correspond to interpolations between the classes (images with hands facing down, with two thumbs are an interpolation between *"dorsal right"* and *"dorsal left"*), while others do not (5 fingers without a thumb). Dynamic guidance is effective in mitigating both.

## 4.2 SCORE FUNCTION SHARPENING

We find that sampling with Dynamic Guidance mitigates hallucinations because it selectively sharpens the score function learned by the diffusion model across the directions associated with the classifier. In the simple 2D Gaussian setup we calculate the unguided score function $s_\theta(\boldsymbol{x}_t, t) = -\frac{\epsilon_\theta(\boldsymbol{x}_t, t)}{\sqrt{1-\bar{\alpha}}}$ and the guided score function $s_\theta^g(\boldsymbol{x}_t, t) = s_\theta(\boldsymbol{x}_t, t) + \lambda \nabla_{\boldsymbol{x}_t} \log p(y^* | \boldsymbol{x}_t)$ and plot them together with the real score function in Figure 3. We see that Dynamic Guidance effectively sharpens the score function, which mitigates hallucinations as shown in Section 5.

For more complex datasets, we want to visualize the learned function for specific directions that have meaningful semantics. We use the Single Shapes dataset, for which we will visually examine the score function learned by a diffusion model and the score function under our proposed Dynamic Guidance. We focus on latent directions that correspond to changing the appearance of a shape, since those variations can lead to hallucinations.

We first learn how to embed images from the dataset using a $\beta$-VAE (Higgins et al., 2017; Burgess et al., 2018) with a disentangled 10-dimensional latent representation. Examining the VAE-learned representations, we identify those that affect specific properties of the image: Dimension 9 controls the appearance of shapes on the left side, and Dimension 5 controls the position of shapes on the right (Figures 1, A.5).

The trained $\beta$-VAE provides a transformation between latents and clean images using the learned encoder $\boldsymbol{z} = \mathcal{E}(\boldsymbol{x}_0)$ and decoder $\boldsymbol{x} = \mathcal{D}(\boldsymbol{z})$. Our goal is to estimate the score of the latents $\nabla_{\boldsymbol{z}} \log p(\boldsymbol{z})$. Using the change of variables formula for this autoencoder setting ((Köthe, 2023)), we can express the distribution of the latents as

$$p_Z(\boldsymbol{z}) = p_{X_0}(\mathcal{D}(\boldsymbol{z}))\sqrt{|\det\left(\boldsymbol{J}_\mathcal{D}^T(\boldsymbol{z})\boldsymbol{J}_\mathcal{D}(\boldsymbol{z})\right)|}, \quad (6)$$

where $\boldsymbol{J}_\mathcal{D}(\boldsymbol{z})$ is the Jacobian of the VAE decoder at $\boldsymbol{z}$. Taking the $\nabla_{\boldsymbol{z}} \log$ on both sides we have

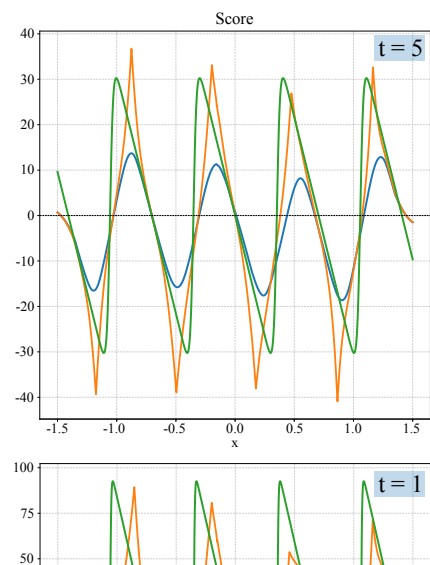

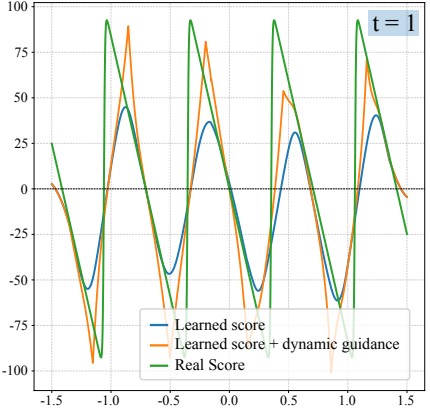

Figure 3: **Score Function Sharpening.** The learned score function of the diffusion model with and without Dynamic Guidance, compared to the true score function for a 2D mixture of Gaussians across the x dimension. The model learns a smoothed-out score function, which Dynamic Guidance sharpens so that it more closely approximates the correct one.

$$\nabla_{\boldsymbol{z}} \log p_Z(\boldsymbol{z}) = \nabla_{\boldsymbol{z}} \log p_{X_0}(\mathcal{D}(\boldsymbol{z})) + \nabla_{\boldsymbol{z}} \log \sqrt{|\det\left(\boldsymbol{J}_\mathcal{D}^T(\boldsymbol{z})\boldsymbol{J}_\mathcal{D}(\boldsymbol{z})\right)|}. \quad (7)$$

To estimate the score of the latents $\nabla_{\boldsymbol{z}} \log p_Z(\boldsymbol{z})$, we compute $\nabla_{\boldsymbol{z}} \log \sqrt{|\det\left(\boldsymbol{J}_\mathcal{D}(\boldsymbol{z})^T \boldsymbol{J}_\mathcal{D}(\boldsymbol{z})\right)|}$ using automatic differentiation. To calculate the score of the clean samples $\nabla_{\boldsymbol{z}} \log p_{X_0}(\mathcal{D}(\boldsymbol{z}))$ we first rewrite it as

$$\nabla_{\boldsymbol{z}} \log p_{X_0}(\mathcal{D}(\boldsymbol{z})) = \boldsymbol{J}_\mathcal{D}^T(\boldsymbol{z}) \nabla_{\boldsymbol{x}_0} \log p_{X_0}(\boldsymbol{x}_0), \quad (8)$$

and then we utilize the Perturb-and-Average Scoring from Wang et al. (2023), which estimates the score of the clean images with an expectation over noisy image scores

$$\nabla_{\boldsymbol{x}_0} \log p_{X_0}(\boldsymbol{x}_0) \approx \mathbb{E}_\epsilon \left[ \nabla_{\boldsymbol{x}_t} \log p_{X_t}(\sqrt{\alpha_t}\boldsymbol{x}_0 + \sqrt{1-\alpha_t}\epsilon) \right]. \quad (9)$$

In practice, we first decode a latent $\boldsymbol{z}$ into a clean image $\boldsymbol{x}_0$. We then perturb $\boldsymbol{x}_0$ with multiple random noise samples at timestep $t$ to get an approximation of the score of $\boldsymbol{z}$ by averaging out the individual predicted scores.

To showcase how the score function is selectively sharpened, we select a generated image $\hat{\boldsymbol{x}}_0$ that is a hallucination, showing a pentagon on the left and a triangle on the right. We plot the score

function given by Equation (7) along the latent dimension 9 (Figure 1). In this instance, the ideal score should be 0 in the regions that correspond to images with a single or two triangles. In contrast, the score for an image containing two shape types should be high, pushing the sample away from the hallucination.

Indeed, this is the exact observation we make in Figure 1; while the base diffusion-learned score does not push the sample away from the hallucination region, Dynamic Guidance sharpens the score predictions, mitigating the hallucination. At the same time, when plotting the score across a latent that is not related to hallucinations and is not captured by the classifier (Latent Dimension 5), Dynamic Guidance has no effect over it.

## 5 REDUCING HALLUCINATIONS WITH DYNAMIC GUIDANCE

### 5.1 CONTROLLED SETTINGS

For all datasets we perform experiments for, we measure the reduction in hallucinations as follows:

$$\text{HR} = \frac{\#\text{Hallucinations}_{\text{before}} - \#\text{Hallucinations}_{\text{after}}}{\#\text{Hallucinations}_{\text{before}}} \quad (10)$$

Negative values correspond to an increase in hallucinations. For the Mixture of 2D Gaussians dataset, we train a simple MLP for both the diffusion model and classifier, while for all controlled image datasets, we train an ADM (Dhariwal & Nichol, 2021) and a noisy sample classifier using the guided-diffusion[1] codebase. We compare to variance filtering, the detection-based method proposed by Aithal et al. (2024), and Classifier Guidance. We pick the best hyperparameters for variance filtering by performing a grid search around the values mentioned in their paper.

Table 1: Performance on Hallucination Reduction on the 2D Mixture of Gaussians dataset.

| Method | # Training iterations | | |
|---|---|---|---|
| | 20k | 50k | 100k |
| Variance Filtering (1%) | 6.1% | 12.71% | 17.8% |
| Variance Filtering (2.5%) | 15.4% | 34.5% | 45.7% |
| Variance Filtering (5%) | 34.5% | 60.5% | 71.3% |
| **Dynamic Guidance** | **69.5%** | **76.1%** | **72.1%** |

Table 2: Performance on Single Shapes.

| Method | HR↑ |
|---|---|
| Variance Filtering (1%) | 1.05% |
| Variance Filtering (5%) | 3.16% |
| Variance Filtering (10%) | 11.58% |
| Classifier Guidance | -33% |
| **Dynamic Guidance** | **74.21%** |

Table 3: Performance of Classifier (CG) and Dynamic Guidance (DG) on Hallucination-related metrics on ImageNet-1k generation.

| | Inception | | | CLIP | |
|---|---|---|---|---|---|
| Method | IS↑ | Prec↑ | Rec↑ | Prec↑ | Rec↑ |
| Uncond. ADM | 34.24 | 0.53 | 0.61 | 0.60 | 0.26 |
| + CG $\lambda = 1$ | 28.52 | 0.47 | 0.62 | 0.57 | 0.25 |
| **+ DG** $\lambda = 1$ | 49.63 | 0.62 | 0.59 | 0.66 | **0.27** |
| + CG $\lambda = 2$ | 38.39 | 0.55 | **0.64** | 0.63 | **0.27** |
| **+ DG** $\lambda = 2$ | 59.31 | 0.70 | 0.56 | 0.70 | 0.26 |
| + CG $\lambda = 10$ | 83.19 | 0.69 | 0.55 | 0.74 | **0.27** |
| **+ DG** $\lambda = 10$ | **88.49** | **0.77** | 0.52 | **0.77** | 0.26 |
| + CG $\lambda = 20$ | 75.84 | 0.62 | 0.55 | 0.70 | 0.23 |
| **+ DG** $\lambda = 20$ | 80.99 | 0.70 | 0.52 | 0.75 | 0.23 |

Table 4: Performance on Mixed Shapes.

| Method | HR↑ |
|---|---|
| Variance Filtering (1%) | 1.35% |
| Variance Filtering (5%) | 8.11% |
| Variance Filtering (10%) | 17.57% |
| Classifier Guidance | -1.92% |
| **Dynamic Guidance** | **72.97%** |

Table 5: Performance on Hands-11k.

| Method | HR↑ |
|---|---|
| Variance Filtering (1%) | 0% |
| Variance Filtering (5%) | 14.28% |
| Variance Filtering (10%) | 14.28% |
| Classifier Guidance | 21.4% |
| **Dynamic Guidance** | **50%** |

---

[1]https://github.com/openai/guided-diffusion

**2D Mixture of Gaussians**    For the 2D Mixture of Gaussians dataset, as both Aithal et al. (2024) and we observe, the number of training data and iterations greatly affect the number of hallucinations, so we train the DDPM Ho et al. (2020) for 20k, 50k, and 100k iterations and generate 100k samples with DDPM for evaluation (Table 1). For variance filtering, we evaluate the performance when discarding 1%, 2.5%, and 5% of the generated samples. We observe that variance filtering requires discarding a very large percentage of samples to match Dynamic Guidance, while our method outperforms it in most cases.

**Single Shapes - Mixed Shapes**    For both Shape datasets, we generate 50k images to train the diffusion and classifiers, and sample 10k images using 50-step DDIM for evaluation. Hallucinations are quantified by automatically detecting shapes in the generated images using OpenCV. To evaluate variance filtering, we test thresholds that discard 1%, 5%, and 10% of generated samples. Even at the highest threshold (10%), variance filtering fails to reliably identify and remove hallucinations. In contrast, Dynamic Guidance consistently mitigates more than 50% of hallucinations across a broad range of guidance scales, and achieves over a 70% reduction at the optimal scale (Tables 2,4).

**Hands**    For the hands dataset (Afifi, 2019), we train both the diffusion models and the classifiers on the 11k images provided, and sample 100 images using 25-step DDIM for evaluation. For evaluation, we manually label each image as hallucinated or not. To evaluate variance filtering, we test thresholds that discard 1%, 5%, and up to 10% of generated samples. Even at the highest threshold (10%), variance filtering fails to reliably identify and remove hallucinations. In contrast, Dynamic Guidance can mitigate as much as 50% of hallucinations even in the challenging setting of sampling with 25 step DDIM (Table 5).

## 5.2 IMAGENET

Hallucinations in large-scale image benchmarks span a vast range of classes, objects, and scenes, making them nearly impossible to precisely define, let alone detect. With this in mind, we choose ImageNet1k generation as a large-scale benchmark to evaluate our approach. In this setting, hallucinations cannot be strictly defined, so we rely on proxy metrics: precision, recall (Kynkäänniemi et al., 2019), and Inception Score (Salimans et al., 2016). Precision measures the fraction of generated samples that fall inside the support of the estimated real data distribution. We argue that higher precision reflects fewer hallucinations, since hallucinations correspond to samples outside the *true* data distribution. Inception Score is maximized when generated images clearly belong to some class (they are low-entropy, high-confidence predictions for the classifier) and diverse (labels evenly distributed). When generating samples, conditioning and initial noise are often mismatched (Li et al., 2025) and images fail to correspond to any valid class, visually resembling hallucinations (Figure 4). A large presence of such samples reduces the classifier's confidence and consequently hurts both the Inception Score and Precision.

In our experiments, we evaluate precision and recall using both the Inception (Szegedy et al., 2016) and CLIP (Radford et al., 2021) models and Inception Score. We adopt the pretrained diffusion model and classifier from Dhariwal & Nichol (2021) and compare three settings: unguided generation, classifier guidance with varying guidance scales, and Dynamic Guidance with the same scales. The results, summarized in Table 3, show that Dynamic Guidance consistently outperforms classifier guidance across all settings. Notably, in the best setting, it achieves precision and Inception Score gains of 8 and 5 points, respectively, while maintaining diversity; recall is not significantly reduced, and the Inception Score is the highest overall. This trend holds across different metrics; we also measure FID with both Inception and CLIP and generative density and coverage (Naeem et al., 2020) and report the results in Tables 8 and 9 of Appendix Section A.2. In Appendix Figures A.11, A.12, we show how 'static' classifier guidance tends to overshoot in cases where the selected condition does not align with the initial noise sample.

### 5.2.1 SELECTION OF GUIDANCE INTERVAL

We identify that, unlike Classifier Guidance, our proposed Dynamic Guidance (DG) can be performed just for a subset of timesteps, which we denote in our algorithm as $[T_1, T_2]$. Classifier Guidance attempts to impose a **strong constraint** on the generation process: the final sample should belong to the chosen class. On the other hand, Dynamic Guidance is more similar to Classifer-Free

Table 6: Ablation for the timestep interval $[T_1, T_2]$ to perform Dynamic Guidance in ImageNet.

| Method | IS↑ | Prec↑ | Rec↑ |
|---|---|---|---|
| Uncond. ADM | 34.24 | 0.53 | 0.61 |
| + DG [1000-0] | 48.20 | 0.73 | 0.33 |
| + DG [600-200] | 61.07 | 0.64 | 0.62 |
| **+ DG [800-400]** | 88.49 | 0.77 | 0.52 |

Table 7: Dynamic Guidance with a classifier trained on DINOv2 pseudo-classes performs as well as DG with ImageNet classes.

| Method | Inception | | | CLIP | |
|---|---|---|---|---|---|
| | IS↑ | Prec↑ | Rec↑ | Prec↑ | Rec↑ |
| Uncond. ADM | 34.24 | 0.53 | 0.61 | 0.60 | 0.26 |
| + CG w/ real | 83.19 | 0.69 | 0.55 | 0.74 | 0.27 |
| + DG w/ real | 88.49 | 0.77 | 0.52 | 0.77 | 0.26 |
| **+ DG w/ clusters** | 75.17 | 0.78 | 0.51 | 0.77 | 0.25 |

Guidance (CFG) in that it attempts to guide a strong signal (conditional model in the case of CFG, unconditional model in the case of DG) with a weak guidance signal (unconditional model in the case of CFG, dynamic gradient of a classifier in the case of DG) to improve generation.

Inspired by recent work on CFG (Kynkäänniemi et al., 2024; Wang et al., 2024), we show that applying DG only for some intermediate timesteps $[T_1, T_2]$ improves performance. To select $T_1$ and $T_2$ for each experiment, we chose the generation timestep at which the image begins to form ($T_1$), and the timestep where samples appear to have already converged to an image that cannot be modified further ($T_2$). We verify our choice of $[T_1, T_2]$ in Table 6, where for our ImageNet experiments the selected $T_1 = 800$ and $T_2 = 400$ achieve the best results. In Figures A.6 and A.7 in the Appendix, we visualize the generation process to show how our choice of $T_1$ and $T_2$ is motivated by the image formation process.

### 5.2.2 GUIDANCE WITH PSEUDO-CLASSES

In the Mixed Shapes experiments, we discussed how Dynamic Guidance can be used with a proxy classifier, whose labels do not necessarily correspond to the hallucinations we aim to avoid. The set of labels we use for the purpose of reducing hallucinations with DG does not have to be identical to the set of labels used to control a generative model. The labels used for conditioning must correspond to human-interpretable semantics, such as object categories, attributes, or text, since they determine what the model is intended to generate. In contrast, the labels used for hallucination reduction do not need to carry such semantic meaning; they may correspond to abstract latent modes or auxiliary partitions of the data that help isolate hallucination-prone directions without mapping to interpretable concepts.

For our ImageNet experiments, we create pseudo-classes by clustering the training images using DINOv2(Oquab et al., 2024). We create 5000 clusters using the hierarchical clustering method described by Vo et al. (2024), assign pseudo-labels to the clusters, and train a classifier to predict those pseudo-labels. In Table 7, Dynamic Guidance with pseudo-classes still improves the hallucination-related metrics, even without ever having access to a set of labels.

## 6 LIMITATIONS

While Dynamic Guidance effectively mitigates hallucinations , it can also impact the diversity of generated samples, as it also introduces certain biases. The impact of Dynamic Guidance on the diversity-hallucination trade-off depends on how the selected classes relate to the hallucination direction. When the chosen classes align with hallucination-relevant directions (e.g., Single Shapes), Dynamic Guidance sharpens the score only where necessary, preserving diversity. When this alignment is not possible, it still mitigates hallucinations but may slightly reduce diversity, as suggested by the modest recall drop and increased coverage on ImageNet. Since the method relies on a classifier to determine the most likely mode at each timestep, any bias in the classifier's predictions can affect the sampling trajectory, leading to a preference for particular classes that are more confidently classified or easier to identify. Consequently, certain semantic modes can receive disproportionately more probability mass, leading to uneven final class distributions. In addition, the initial noise introduces bias in the shape or position of objects in the generated image, which does not fit all classes

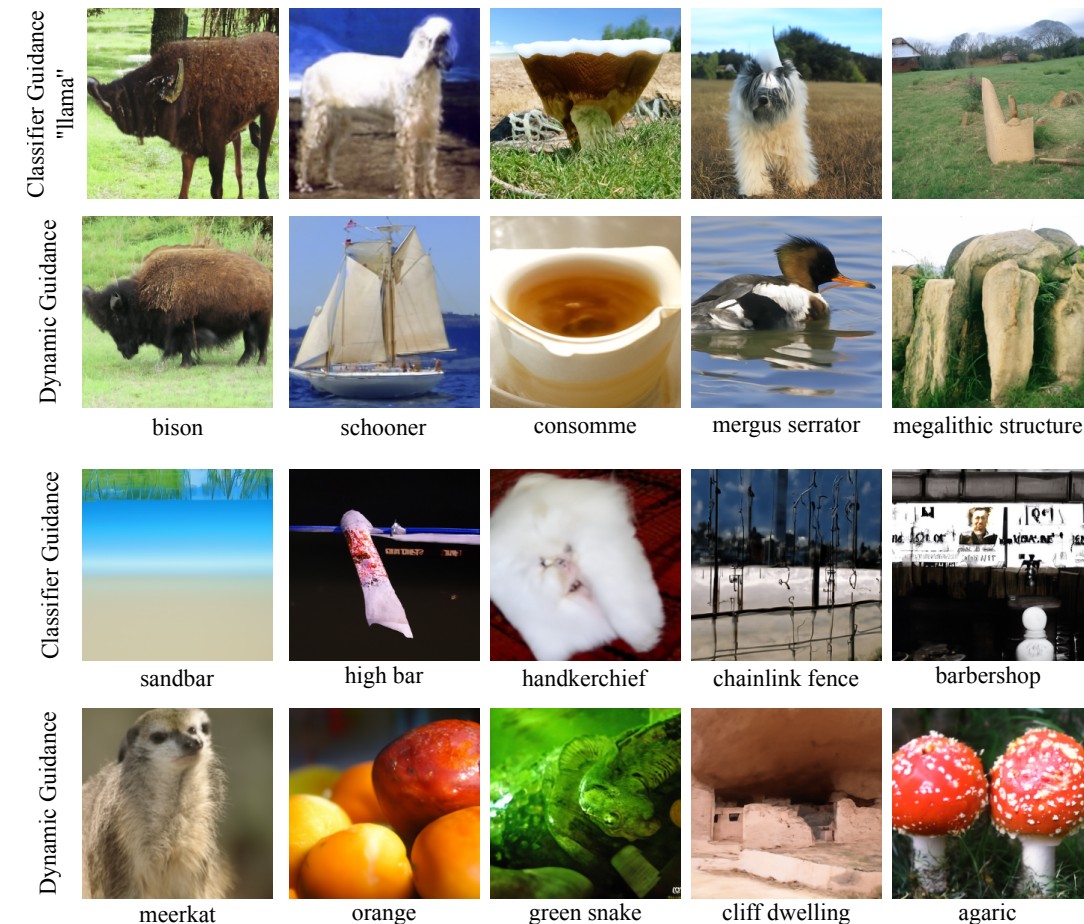

Figure 4: Images generated with Classifier and Dynamic Guidance using the same initial noises. **(Top)** For a specific label ("llama") that is sometimes misaligned with the initial noise, the diffusion model can generate low-quality samples that visually resemble hallucinations. **(Bottom)** Even for randomly selected labels, when those are fixed, misalignment can produce generations that would be considered hallucinations.

uniformly, further skewing the distribution of generated samples. In practice, we find that certain classes are over-represented (Figures A.17 and A.18) when using Dynamic Guidance.

## 7 CONCLUSION

In this work, we addressed the problem of hallucinations in diffusion models by introducing Dynamic Guidance, a method that mitigates hallucinations during the generative process itself. Unlike prior detection-based approaches, Dynamic Guidance prevents hallucinations from arising by selectively sharpening the score function along hallucination-inducing directions while preserving benign interpolations that support diversity. Our experiments across toy data, controlled and real-world image datasets demonstrate consistent improvements, with Dynamic Guidance achieving substantial hallucination reduction even under realistic low-step DDIM sampling. On the large-scale benchmark ImageNet, we further show improvements in proxy metrics such as precision and Inception Score, validating that our method generates samples that remain closer to the true data distribution. We believe Dynamic Guidance provides a principled step toward more reliable diffusion models and opens opportunities for future work in understanding and controlling hallucinations in large-scale generative models.

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

# A APPENDIX

## A.1 DYNAMIC GUIDANCE WITH DDPM

Extending Dynamic Guidance to DDPM is straightforward. We describe the algorithm in Algorithm 2. In the main paper, we employed DDIM as it is the more practical sampling algorithm for models larger than the toy 2D Gaussian setting.

---

**Algorithm 2** Dynamic Guidance with DDPM

**Input:** timesteps $T$, dynamic guidance steps $T_1$ $T_2$, denoiser $\mu\theta$, classifier $p_\phi$
$\boldsymbol{x}_T \sim \mathcal{N}(0, \mathbf{I})$
**for** $t = T \dots 1$ **do**
    $\mathbf{z} \sim \mathcal{N}(0, \mathbf{I})$
    **if** $T_1 \geq t \geq T_2$ **then**
        $y^* = \arg\max p_\phi(y|\boldsymbol{x}_t)$
        $\boldsymbol{x}_{t-1} = \mu_\theta(\boldsymbol{x}_t, t) + \sigma_t^2 \lambda \nabla_{\boldsymbol{x}_t} p_\phi(y^*|\boldsymbol{x}_t) + \sigma_t \mathbf{z}$
    **else**
        $\boldsymbol{x}_{t-1} = \mu_\theta(\boldsymbol{x}_t, t) + \sigma_t \mathbf{z}$
    **end if**
**end for**
**return** $\boldsymbol{x}_0$

---

## A.2 ADDITIONAL METRICS

We also compute Inception and CLIP FID and observe that in most settings Dynamic Guidance outperforms Classifier Guidance. For high guidance scales ($\lambda = 10$), Dynamic Guidance tends to generate classes that are better matched with a wider range of initial noises, and so it ends up creating a distribution of generated samples that is non-uniform with regard to the ImageNet classes. This greatly affects Inception FID since the Inception model is heavily biased towards balanced ImageNet generation (trained on ImageNet-1k), whereas performance on CLIP FID is not affected. To fairly compare Dynamic Guidance to Classifier Guidance that strictly enforces a balanced generated distribution for evaluation, we generate a larger amount of samples and perform stratified sampling on the generated set to approximate a balanced distribution. We report the results in Table 8 in the row titled *+ DG $\lambda = 10$ (balanced)*.

Table 8: Inception and CLIP FID on ImageNet.

| Method | Inception FID↓ | CLIP FID↓ |
|---|---|---|
| Uncond. ADM | 37.48 | 42.93 |
| + CG $\lambda = 1$ | 43.72 | 48.02 |
| + DG $\lambda = 1$ | 27.46 | 38.21 |
| + CG $\lambda = 2$ | 32.96 | 41.15 |
| + DG $\lambda = 2$ | 25.36 | 36.66 |
| + CG $\lambda = 10$ | 17.05 | 30.01 |
| + DG $\lambda = 10$ | 25.57 | 32.68 |
| **+ DG $\lambda = 10$ (balanced)** | **15.52** | **27.93** |
| + CG $\lambda = 20$ | 21.45 | 40.50 |
| + DG $\lambda = 20$ | 25.58 | 37.07 |

We also report additional metrics aiming to better illustrate the trade-off between reducing hallucinations and loss of diversity in generation. We compute diversity and coverage, as described by Naeem et al. (2020) using CLIP, and report the results in Table 9. We see that DG improves the density of the generations from 0.70 to 0.97 while not sacrificing coverage (0.66 vs 0.67).

Table 9: Density and Coverage on ImageNet-1k generations, based on CLIP.

| Method | Density ↑ | Coverage ↑ |
|--------|-----------|------------|
| Uncond. | 0.7003 | 0.6662 |
| +CG ($\lambda = 1$) | 0.5366 | 0.4473 |
| +DG ($\lambda = 1$) | 0.7023 | 0.5921 |
| +CG ($\lambda = 10$) | 0.9058 | 0.7516 |
| +DG ($\lambda = 10$) | 0.9724 | 0.6761 |
| +CG ($\lambda = 20$) | 0.7999 | 0.6943 |
| +DG ($\lambda = 20$) | 0.9285 | 0.6385 |

### A.3 ADDITIONAL FIGURES

In this section, we provide additional Figures and qualitative examples:

- Figure A.5 shows that change in different latent dimensions learned by the $\beta$-VAE alter the image in distinct; controlled ways.

- Figure A.8 shows how Dynamic Guidance fixes a sample that would end up being a hallucination during sampling.

- Figure A.6 shows how Dynamic Guidance fixes a sample that Classifier Guidance would fail to fix.

- Figure A.7 shows how much the predicted label changes during sampling Dynamic Guidance.

- Figures A.9 and A.10 show 100 images of hands generated with DDIM and Dynamic Guidance, respectively.

- Figures A.11 and A.12 show the difference in the norms of the denoising steps when using Classifier or Dynamic Guidance.

- Figures A.13, A.14, A.15, and A.16 show examples of random images generated with Classifier or Dynamic Guidance with different guidance scales.

- Figures A.17 and A.18 show the final classification for the generated images using Classifier Guidance and Dynamic Guidance.

- Figure A.19 shows that Dynamic Guidance can be applied to a conditional model and improve generations even though the diffusion and guidance signals are not conditioned on the same classes.

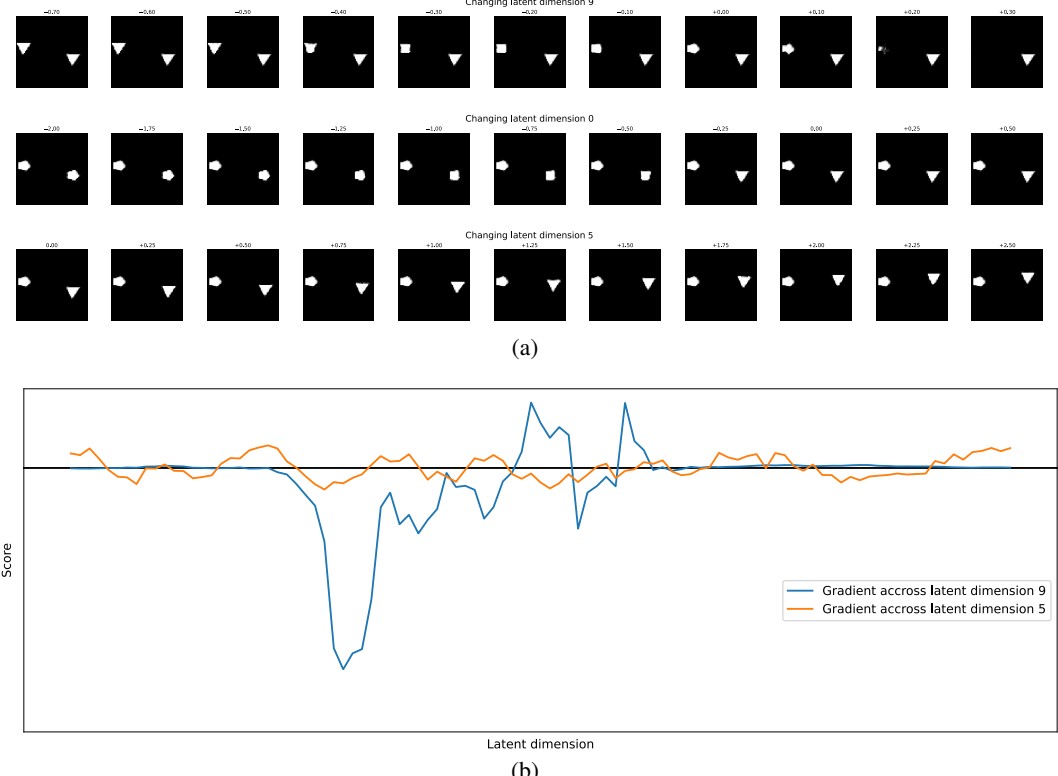

(a)

(b)

Figure A.5: (a) Change in different latent dimensions learned by the $\beta$-VAE alters the image in distinct ways; latent dimension 9 controls the appearance of the shape on the left, latent dimension 0 controls the appearance of the shape on the right, and latent dimension 5 controls the position of the shape on the right. (b) Dynamic Guidance isolates latent dimensions corresponding to hallucinations (dimension 9), while not affecting unrelated ones (dimension 5), as the gradients along hallucination-relevant directions are strong and informative, whereas gradients along irrelevant directions are noisy and close to zero. This is more pronounced when classes are selected so that interpolation between them directly aligns with hallucinations.

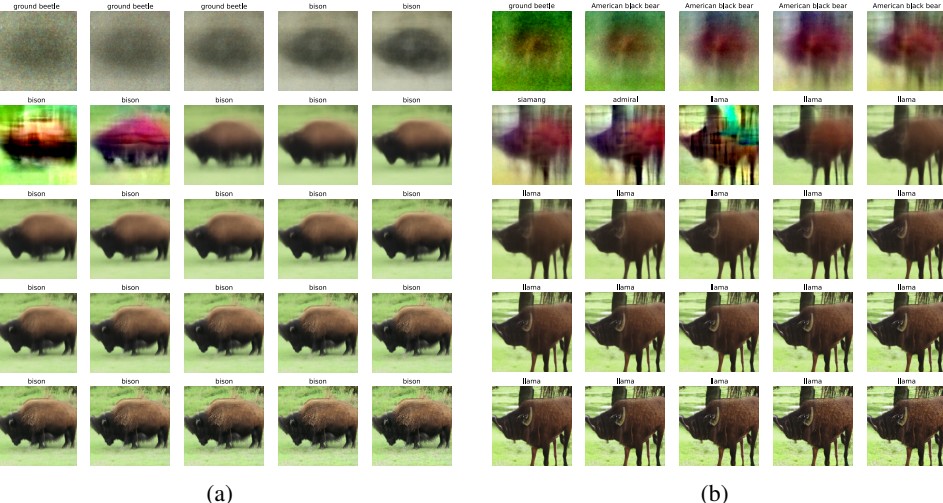

(a)           (b)

Figure A.6: (a) DG steps: When using dynamic guidance, the generated image has the label "bison". The initial noise adds a blob of black pixels in the middle on a green background, for which the local mode is a bison, as predicted by the classifier. By applying the gradients from dynamic guidance, we end up at a realistic-looking bison image, where both the model and classifier agree. (b) CG steps: For the same initial noise, we choose a class that is unlikely to contain a large blob of black pixels in the middle (e.g., "llama"). We see that the classifier guidance gradient updates attempt to correct the image by changing the shape of the generated object and adding thin legs.

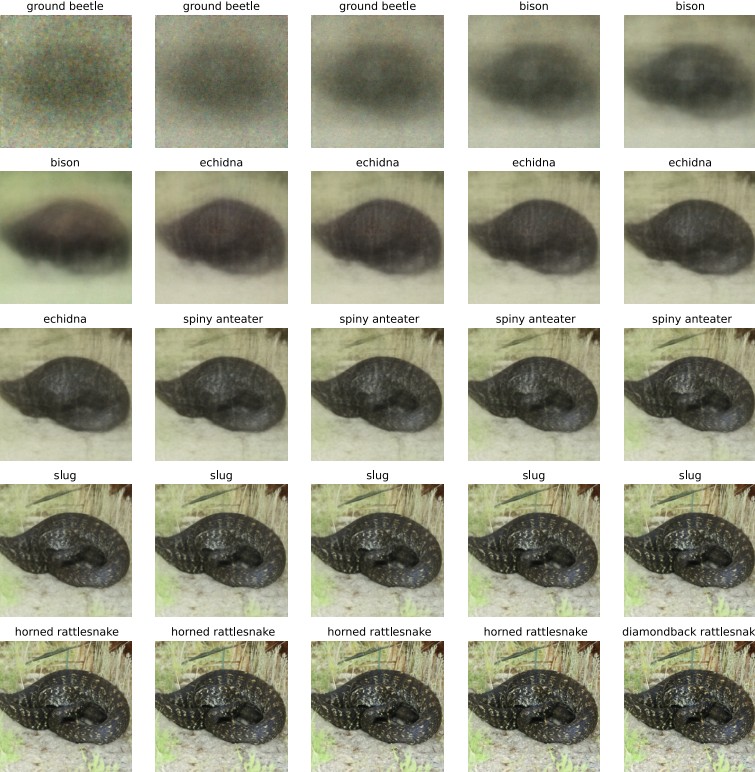

Figure A.7: The intermediate image can be close to multiple modes during generation. The selected mode can change multiple times during sampling, and Dynamic Guidance adaptively chooses the closest one.

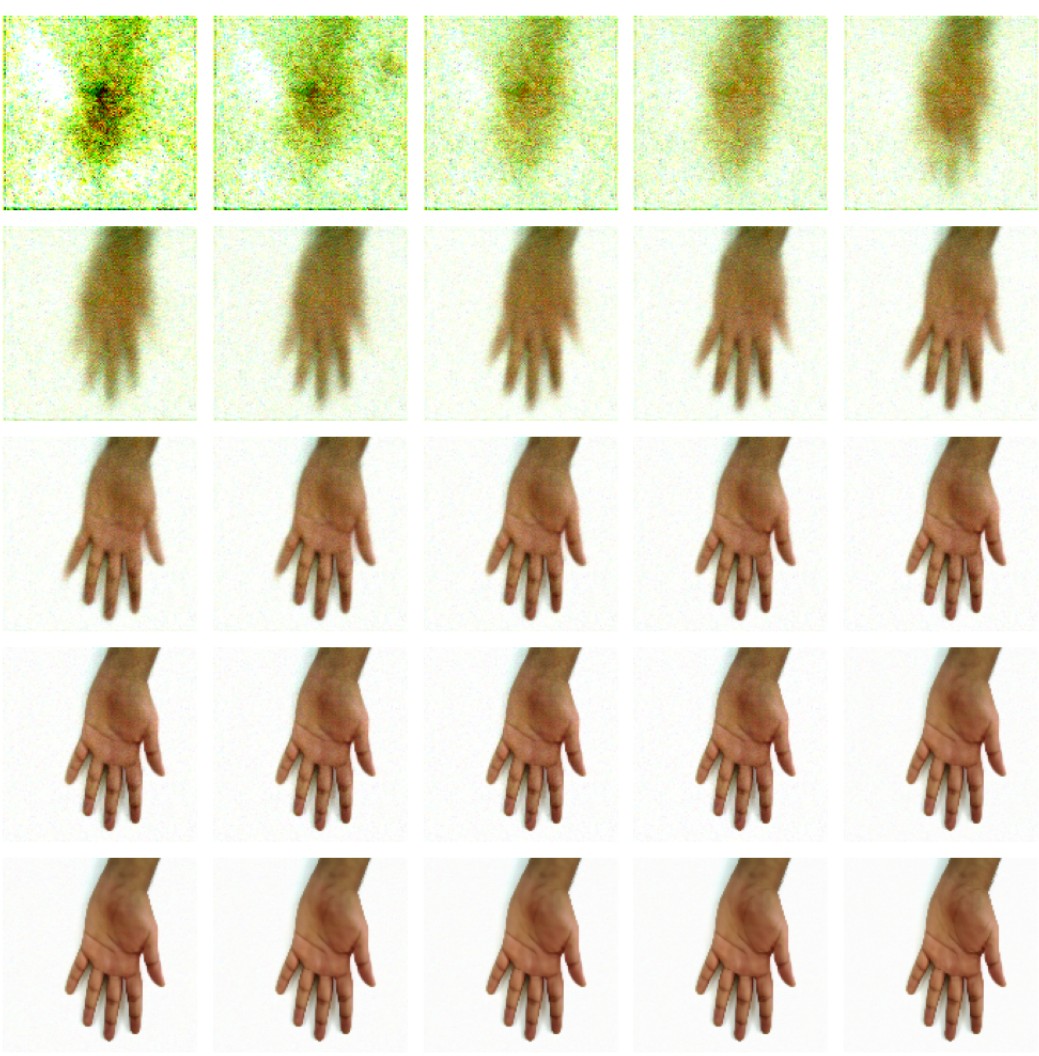

Figure A.8: $\hat{x}_0$ during sampling with Dynamic Guidance using initial noise $x_T$ that would result in a hallucination. We see that Dynamic Guidance guides the model to generate the thumb that would otherwise be missing, resulting in a sample with correct anatomy.

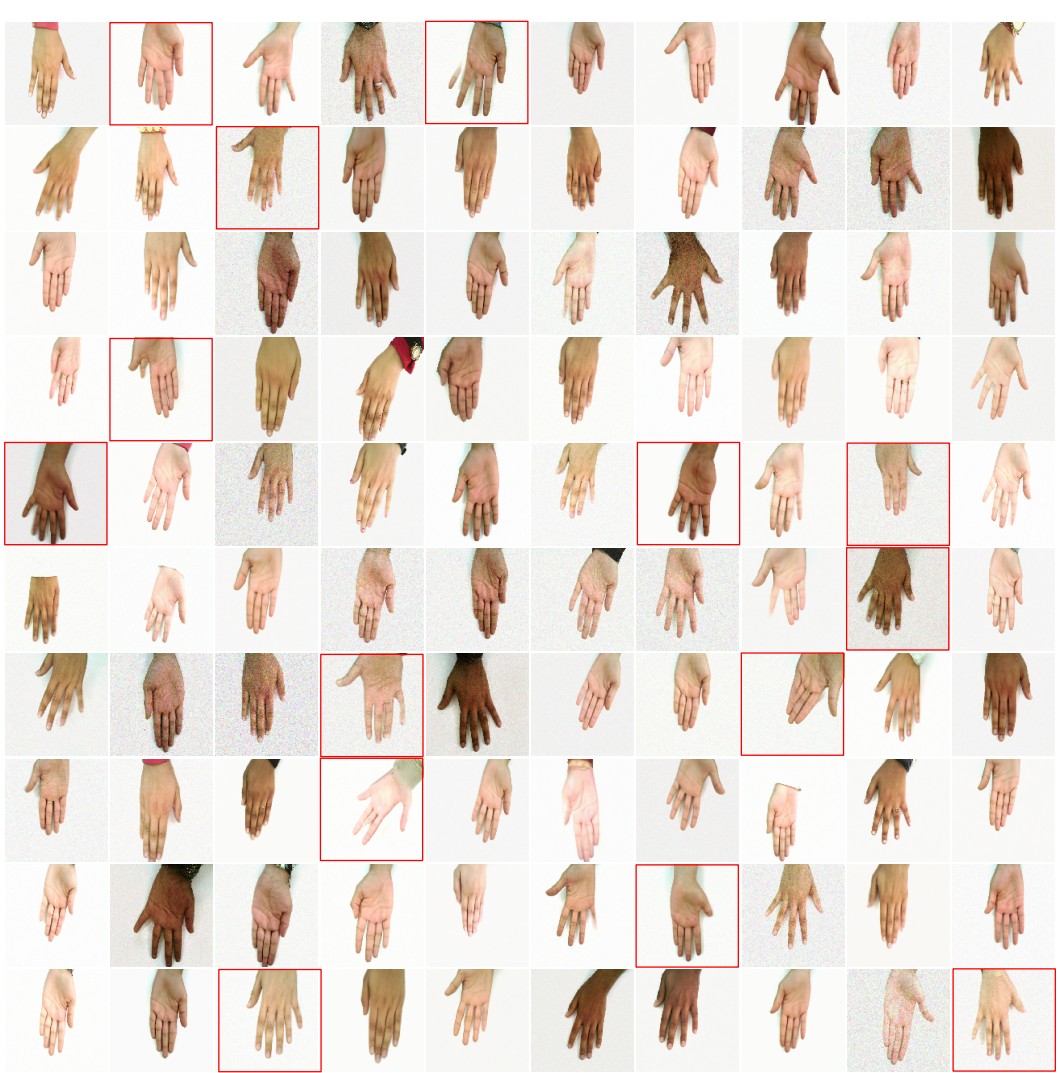

Figure A.9: Generated samples from the hands dataset using DDIM. Hallucinations in red.

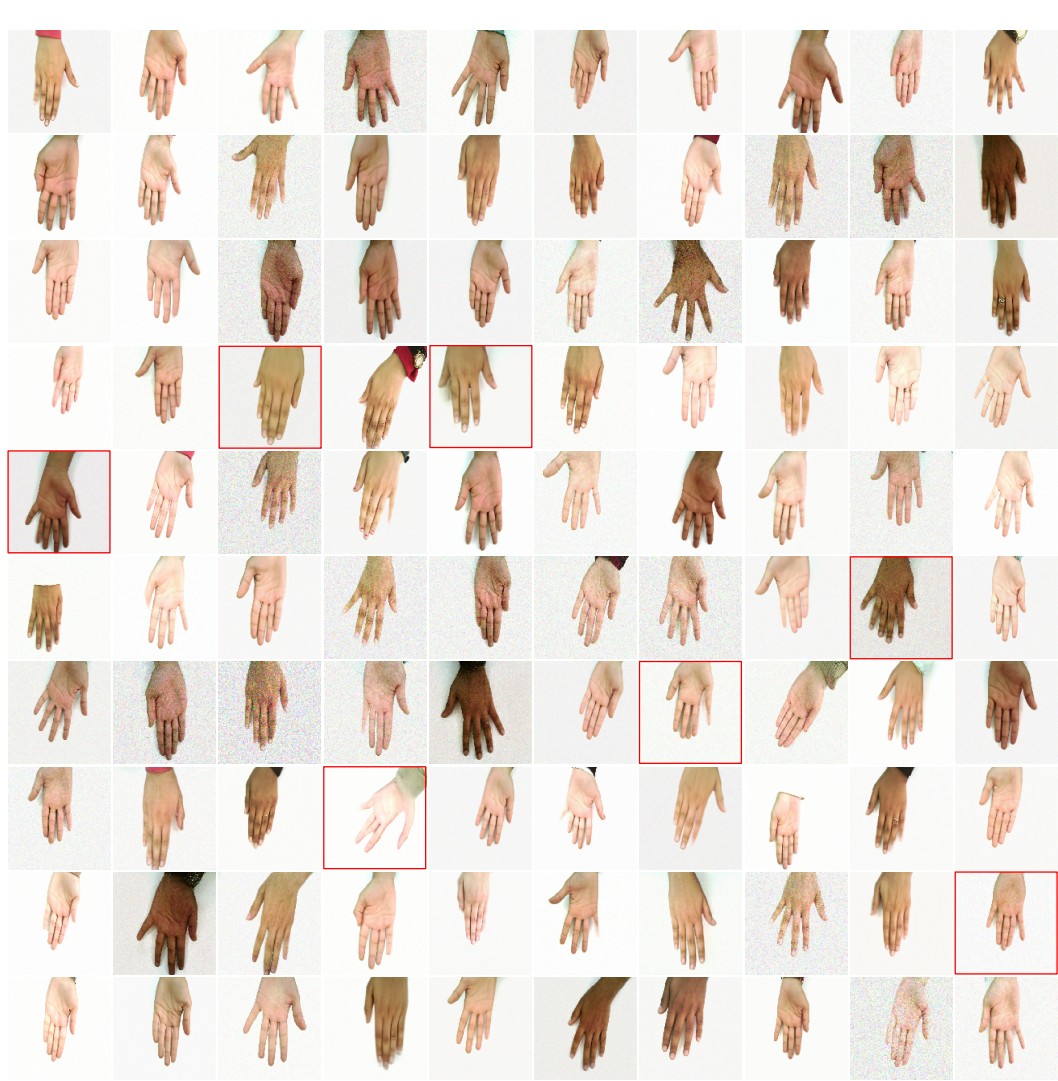

Figure A.10: Generated samples from the hands dataset using Dynamic Guidance. Hallucinations in red.

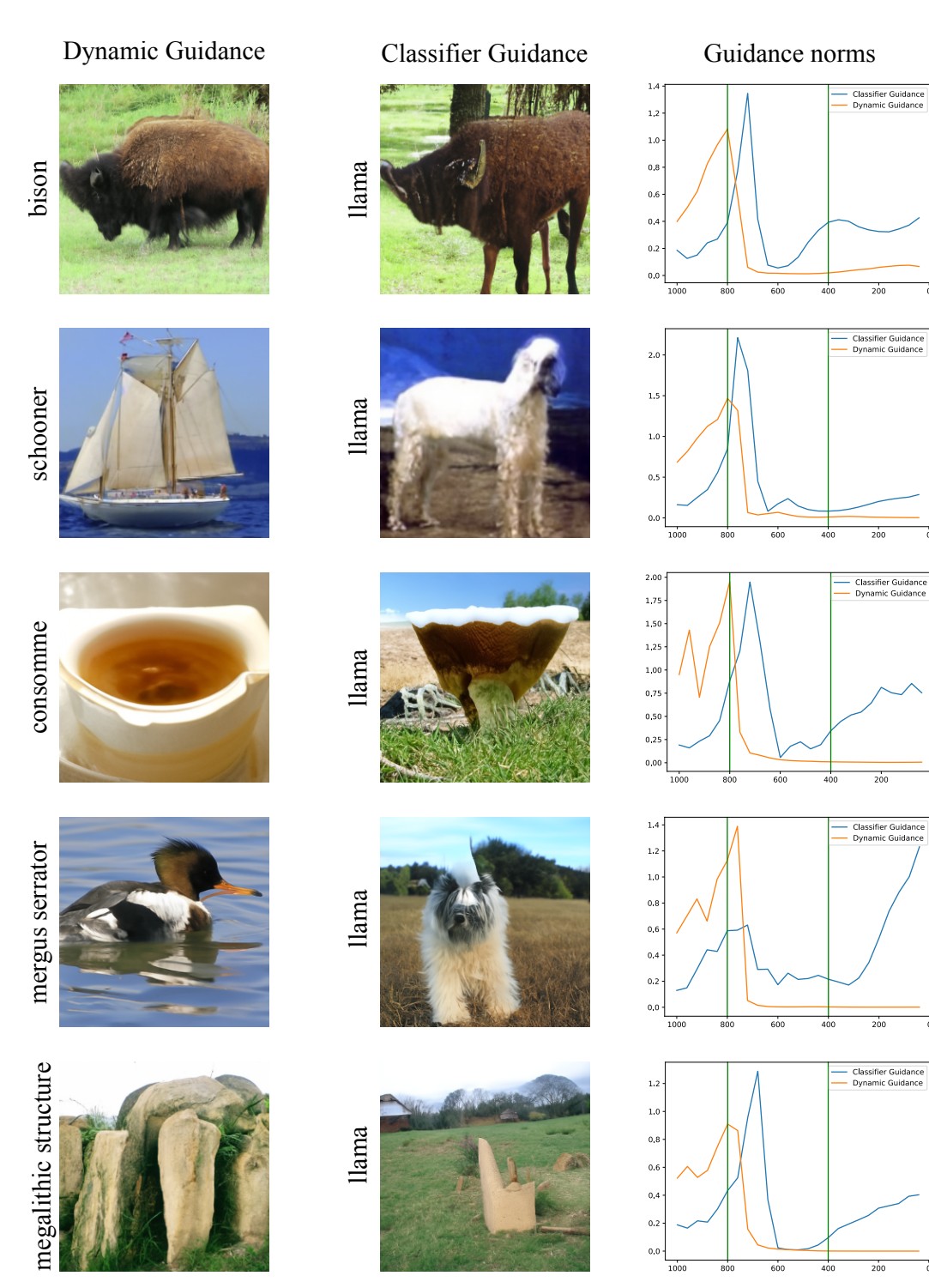

Figure A.11: Generated samples from ImageNet using Classifier Guidance with a fixed label and Dynamic Guidance. In classifier-guided samples, the norm of the denoising step gets bigger towards the end of sampling, meaning that the required denoising steps become significantly larger.

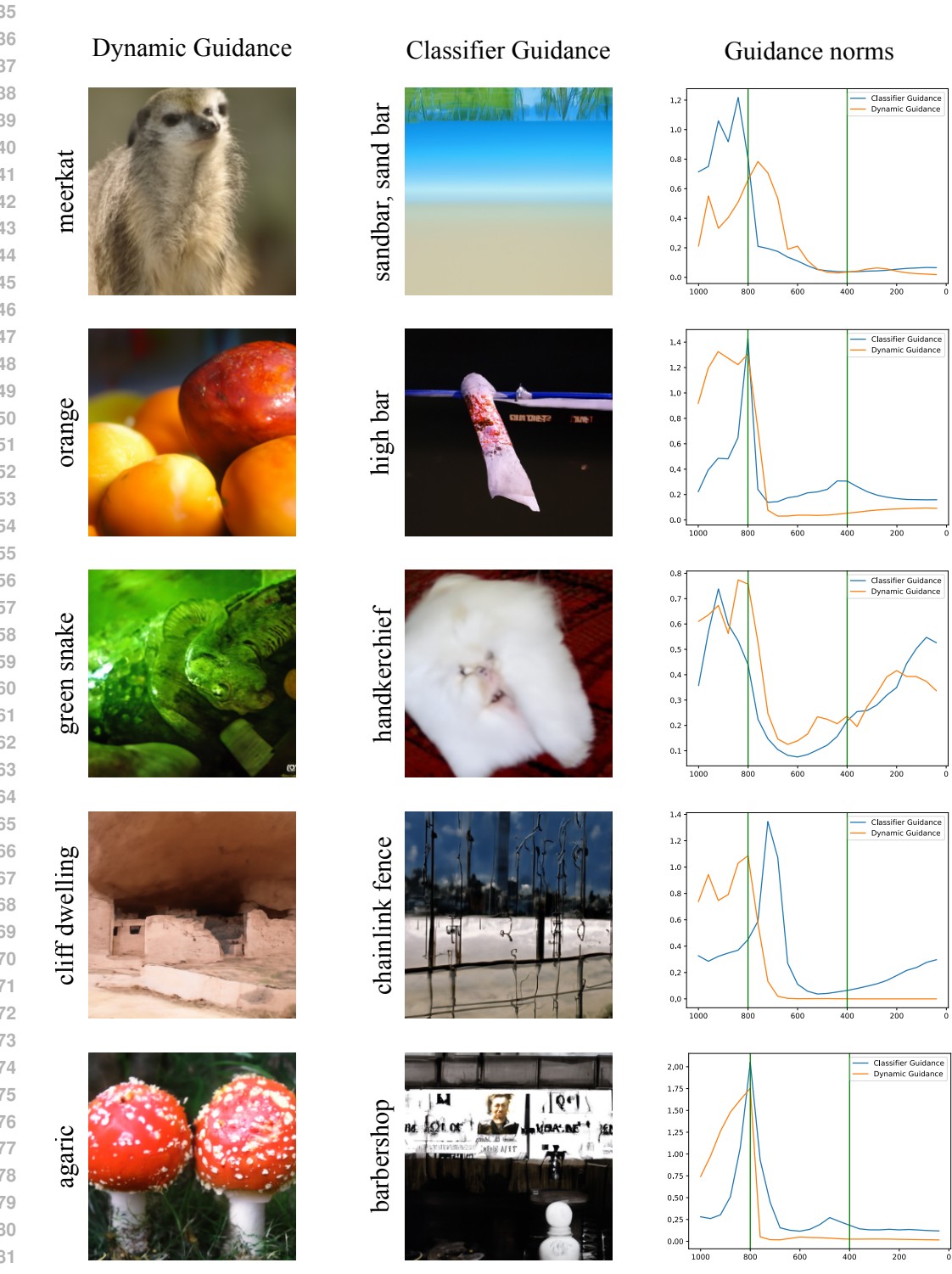

Figure A.12: Generated samples from ImageNet using Classifier Guidance with a random label and Dynamic Guidance. In classifier-guided samples, the norm of the denoising step gets bigger towards the end of sampling, meaning that the required denoising steps become significantly larger.

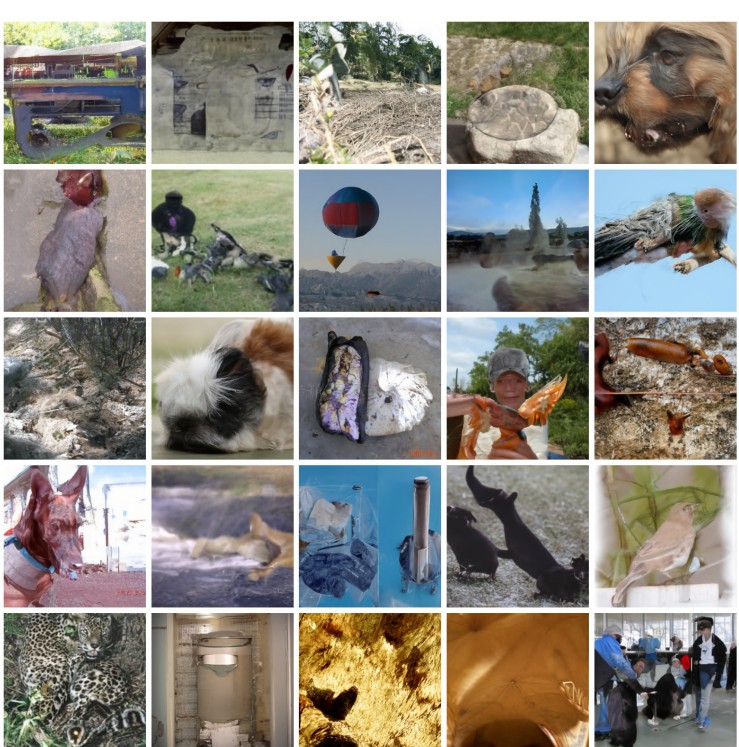

Figure A.13: Random ImageNet samples generated with Classifier Guidance using $\lambda = 1$.

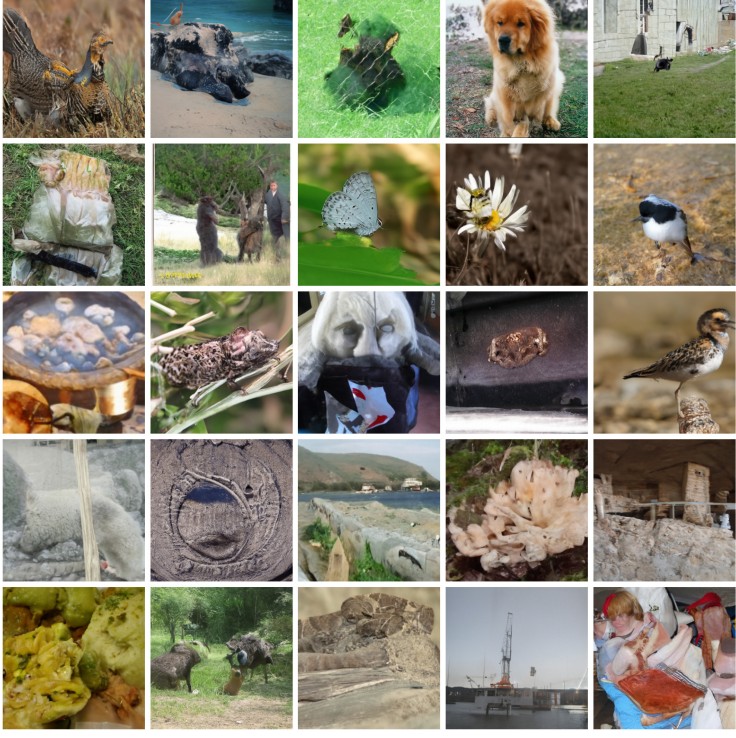

Figure A.14: Random ImageNet samples generated with Dynamic Guidance using $\lambda = 1$.

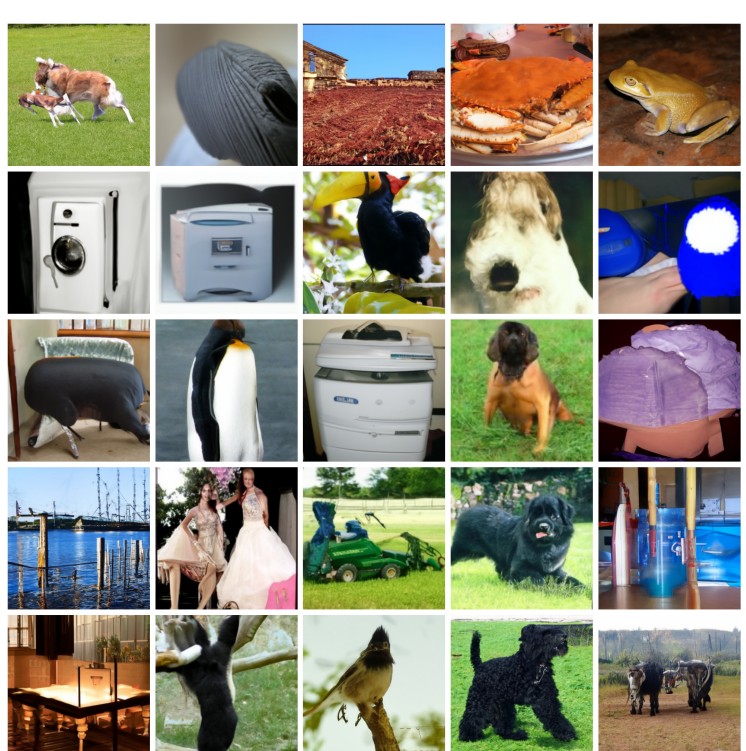

Figure A.15: Random ImageNet samples generated with Classifier Guidance using $\lambda = 10$.

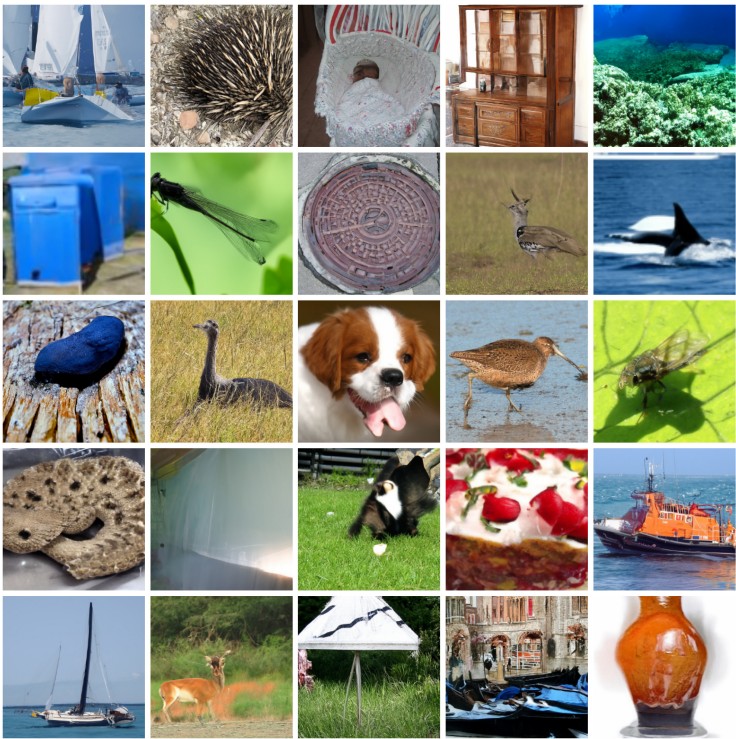

Figure A.16: Random ImageNet samples generated with Dynamic Guidance using $\lambda = 10$.

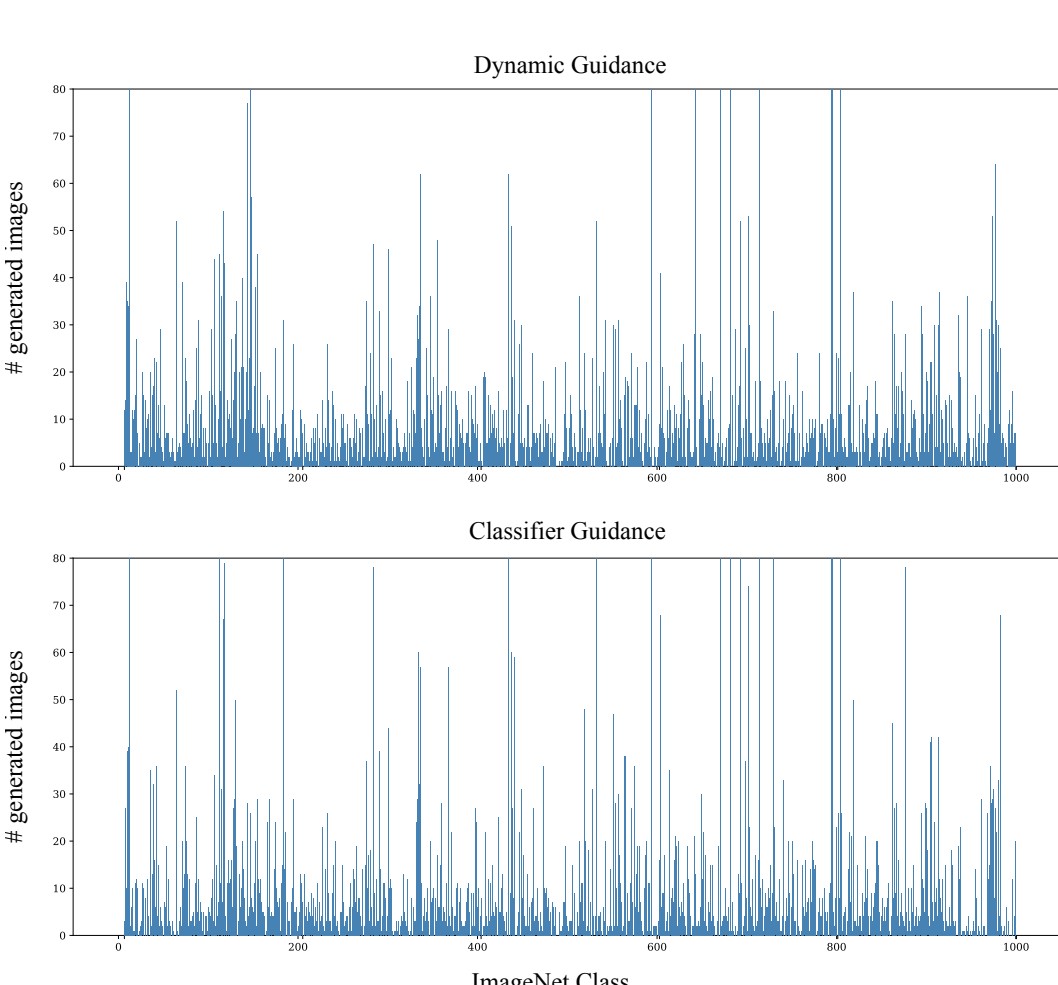

Figure A.17: Distribution of final predicted ImageNet classes for samples generated with Classifier and Dynamic Guidance using $\lambda = 1$.

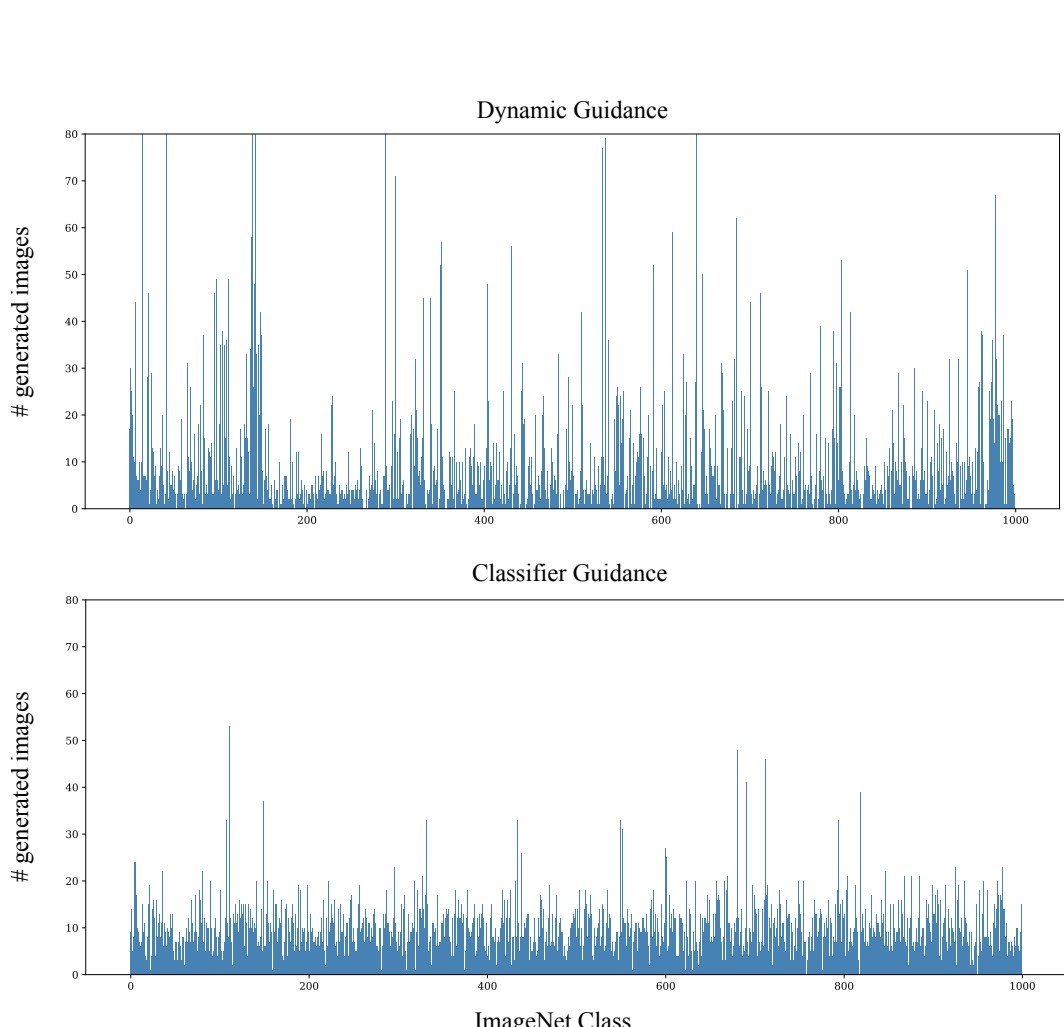

Figure A.18: Distribution of final predicted ImageNet classes for samples generated with Classifier and Dynamic Guidance using $\lambda = 10$.

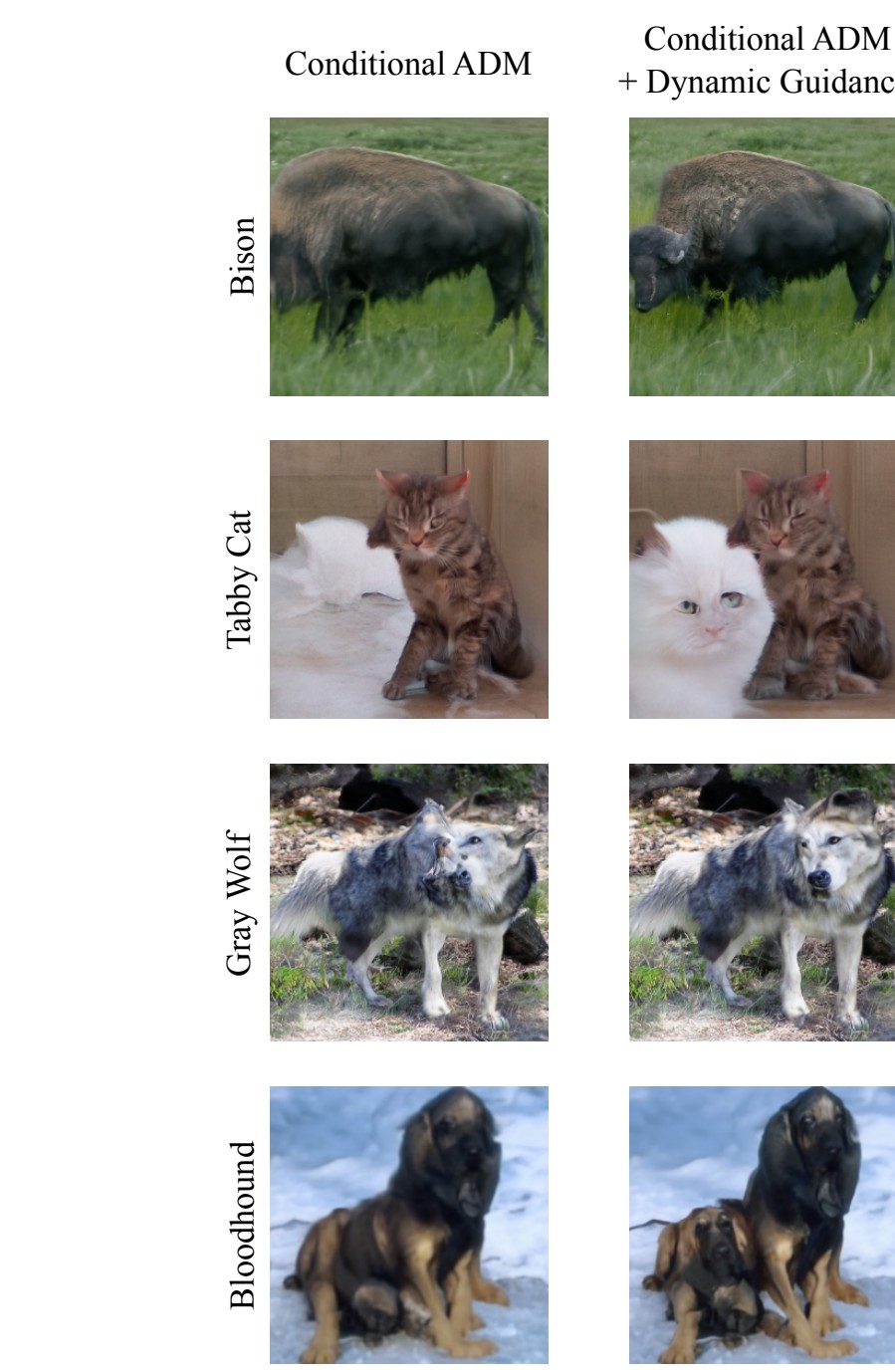

Conditional ADM

Conditional ADM
+ Dynamic Guidance

Bison

Tabby Cat

Gray Wolf

Bloodhound

Figure A.19: Samples generated from an ImageNet class-conditional ADM model, without guidance, and with Dynamic Guidance using the classifier trained on pseudoclasses created by clustering with DINOv2 embeddings (Section 5.2.2). We show that the conditioning and guidance labels do not need to be the same; Dynamic Guidance can improve generations even though the diffusion and guidance signals are not conditioned on the same classes. We practically see that the two can act orthogonally, with class-conditioning guiding the sample towards specific class-related attributes and and dynamic guidance helping avoid bad generations within the class. This is especially apparent in the second and last examples where Dynamic Guidance prevents hallucinations (cat with no face, dog with 5 visible legs) by guiding the samples towards clusters that represent multiple animals in an image.

