# OpenReview forum: "Mitigating Diffusion Model Hallucinations with Dynamic Guidance"
_ICLR.cc/2026/Conference — Submitted to ICLR 2026_

### Official Review · Reviewer_Qih7 · 2025-10-21

**Soundness:** 3
**Presentation:** 3
**Contribution:** 2
**Rating:** 6
**Confidence:** 4

**Summary:**

This paper aims to mitigate hallucinations in diffusion-generated images. The core idea is to use a dynamic classifier guidance scale at each timestep to replace the previous fixed guidance. Such dynamic guidance is done by performing classification on the noisy sample per timestep. Huge and consistent gain is experimentally reported, and the authors also provide detailed empirical analyses on the effects of Dynamic Guidance.

**Strengths:**

1. This work is well-motivated, and the proposed method is closely related to the motivation.
2. The simplicity of the proposed method makes it very easy to integrate into current pipelines.
3. The experimental improvement is impressive, and there are also many empirical observation experiments.

**Weaknesses:**

1. (Major) This work is an extension of classifier guidance, but commercial-scale diffusion models mostly use classifier-free guidance nowadays, where there are uncountable classes, as semantics are continuous. Is there any possibility to extend the idea of Dynamic Guidance to classifier-free guidance?
2. (Minor) While the score sharpening effect is empirically observed with abundant experiments, could there be some more rigorous and theoretical discussion about why this can happen? In particular, why can Dynamic Guidance isolate dimensions related to hallucinations from unrelated ones?
3. (Minor) Language-wise, many commas are missing where there should be a punctuation. A standard grammar check pass would easily fix this.
4. (Minor) Dynamic Guidance can fail if the classifier is not well trained. This is a natural result of being based on classifier guidance, so I won't blame the authors for this weakness.

**Questions:**

The authors mention that there are also studies trying to address hallucinations from the perspective of text-image misalignment. Why do the authors prefer to address this problem at the sampling stage rather than the training stage, which might be more fundamental in my opinion?

---

> ### Author Response · Authors · 2025-11-22
> **Response to reviewer Qih7 (1/2)**
>
> Thank you for your time and thoughtful suggestions, especially with regard to clarifying the theoretical justification for how our proposed Dynamic Guidance reduces interpolations that result in hallucinations while not affecting unrelated ones. Here we address each of your concerns:
>
> ### **1. Connection to CFG**
>
> We emphasize that our work is orthogonal to CFG. CFG improves a conditional model by guiding with an unconditional model. DG improves the model using the gradients of an external classifier. This means that our DG can be used together with classifier-free guidance. Crucially, we also want to emphasize that DG does not require using a set of labels that correspond to actual classes or is identical to the set of labels used to control a generative model with CFG. The labels used for conditioning must correspond to human-interpretable semantics, such as object categories, attributes, or text, since they determine what the model is intended to generate. In contrast, the labels used for hallucination reduction do not need to carry such semantic meaning; they may correspond to abstract latent modes or auxiliary partitions of the data that help isolate hallucination-prone directions without mapping to interpretable concepts. We show that, for ImageNet, we can create pseudo-classes by clustering the training images using DINOv2. We train a classifier on those pseudo-classes. While the selected number of 5000 clusters does not directly correspond to the set of combinatorial subclasses of ImageNet labels, DG using pseudo-subclasses can still improve the model with regard to hallucinations. We have added those results in the main text in Table 7 and created Section 5.2.2 to expand more on this discussion.
>
> | Method                 | IS ↑  | Inception Prec ↑ | Inception Rec ↑ | CLIP Prec ↑ | CLIP Rec ↑ |
> |------------------------|-------|------------------|-----------------|-------------|-------------|
> | **Uncond. ADM**        | 34.24 | 0.53             | 0.61            | 0.60        | 0.26        |
> | **+ CG real classes**  | 83.19 | 0.69             | 0.55            | 0.74        | 0.27    |
> | **+ DG real classes**  | 88.49 | 0.77             | 0.52            | 0.77        | 0.26        |
> | **+ DG dino clusters** | 75.17 | 0.78             | 0.51            | 0.77        | 0.25        |
>
> This means that in theory we can use both our proposed DG and CFG on distinct sets of labels as follows: $(1+w_1)p(x|c) + w_2p(x|c^*) - (w1+w2)p(x)$, where $c^{\*}$ is the dynamic guidance class that belongs in a discrete set, while $c$ can be continuous.
>
> ### **2. Theory behind score sharpening**
>
> Dynamic Guidance relies on the gradient produced by the classifier. If this gradient corresponds to the classifier changing attributes related to the classification label, then we show that the altered score function only focuses on the dimensions where interpolation concerns misclassification (Single Shapes). **In Appendix Figure A.5 we add (b)** that shows that the gradients along hallucination-relevant directions are strong and informative, whereas gradients along irrelevant directions are noisy and close to zero.
>
> Additionally, diffusion models rely on neural networks that inherently represent smooth functions, which limits their ability to model regions where the true score exhibits sharp transitions or discontinuities, such as boundaries between semantic modes. These regions often correspond to hallucination-prone transitions in the data manifold. Dynamic Guidance introduces an external discrete signal (the class indicator), effectively switching between different classifier gradients across modes. We hypothesize that this discrete switching helps approximate sharp transitions in the score function that the diffusion model alone cannot capture due to architectural smoothness constraints. In this view, Dynamic Guidance can compensate for the challenges of representing those discontinuous boundaries by injecting discrete mode-specific directional information that sharpens the score selectively.
>
> ### **3. Grammar and punctuation**
>
> > Language-wise, many commas are missing where there should be a punctuation.
>
>
> Thank you for pointing this out. We have updated the manuscript to fix those.

---

> > ### Comment · Reviewer_Qih7 · 2025-11-24
> >
> > I am partially satisfied with the response. My remaining concerns are:
> > 1. Using pseudo labels rather than real labels to reduce class count feels more like a workaround. Alternatively, the authors could actually apply this method to models like Stable Diffusion to show its practicality.
> > 2. As for the new theoretical explanation, I still feel some gap between it and the true essence. Nevertheless, the addition to the manuscript is welcome.
> > My score was already a positive one. I would keep it unchanged for now and see the other reviewers' opinions.

---

> > > ### Author Response · Authors · 2025-11-26
> > > **Response to reviewer Qih7**
> > >
> > > Thank you for your response.
> > >
> > > > Alternatively, the authors could actually apply this method to models like Stable Diffusion to show its practicality.
> > >
> > > Stable Diffusion requires additional steps (classifiers trained on noisy latents instead of noisy images). As a proof-of-concept we show how DG works with a class-conditioned diffusion model in pixel space: please refer to our response to reviewer yFiB, where we show that the conditioned model and Dynamic Guidance can work together effectively even when using disjoint sets of labels. By showing that DG can improve class-conditioned models, we can sketch an extension to Stable Diffusion, which uses text for conditioning. We leave detailed experimentation with latent diffusion models (which require the DG classifier to operate in the latent space now instead) and how to incorporate DG with classifier-free guidance to future work.

---

> > > > ### Comment · Reviewer_Qih7 · 2025-11-27
> > > >
> > > > I admit the practicality implied by the disjointed label experiment, but it is still indirect evidence.
> > > >
> > > > Still, I think the description of score 6 best suits my feeling: I lean towards acceptance, but wouldn't mind if otherwise. Nevertheless, **I would raise the score to 7 if there were such an option**.
> > > >
> > > > If this paper were accepted, I hope the authors could include a t2i experiment in the camera-ready version or when releasing the codes, instead of leaving it for future work. If it were unfortunately not accepted, I think the authors could further delve deeper into t2i applicability and theoretical insights to make a stronger version.
> > > >
> > > > A final remark to summarize my thoughts:
> > > > 1. The applicability to t2i models is a major concern with more than one reviewer. The authors indeed provide some new experiments suggesting such applicability, but there is no direct evidence yet.
> > > > 2. Since the method is very simple, one could expect some analytic insights from the paper. The authors indeed offer very thorough and abundant empirical analysis, which does provide important insights, but more rigorous and theoretical insights are limited.

---

> ### Author Response · Authors · 2025-11-22
> **Response to reviewer Qih7 (2/2)**
>
> ### **4. What happens if classifier is not well trained?**
>
> In this work, our main argument is that the key to mitigating hallucinations is the classifier-produced gradient that we add at each sampling step, and how it can help with pushing the generated samples away from regions we consider hallucinations. If the classifier is not well-trained, we can either expect it to give uninformative (noisy) gradients that could either have minimal effect on the generated image or badly bias the generation steps. For the classifiers we use in our experiments (ImageNet classes, DINOv2 clustering-based pseudo-classes), training is not a big issue.
>
> Overall, we would like to discuss that in this work, we focus on the effect of the classifier gradient on sampling (Section 4.2, Figure 3), and how it helps with hallucination mitigation (Section 5). We believe that our idea of dynamically choosing the classification target can be applied without having to explicitly train classifiers for the data, for example, by using VLMs for which we dynamically change the target text as we generate the image.
>
> ### **5. Addressing hallucinations during sampling vs training**
> To address this, we use the 2D Gaussian setting from our paper. We train models on the same set of 50k samples for a different number of iterations and report the hallucination rate of the base models.
>
> | Training Iterations | Hallucination Rate (%) |
> |--------------------:|------------------------:|
> | **5k**   | 11.5 ± 0.91 |
> | **10k**  | 2.6 ± 0.42  |
> | **50k**  | 0.28 ± 0.05 |
> | **100k** | 0.10 ± 0.02 |
>
> We show that improving the training of the generative model can reduce hallucinations, as the learned score becomes a sharper and more accurate approximation of the true score. However, the training regime of modern large-scale diffusion models typically resembles the undertrained case: these models are trained on massive datasets and rarely, if ever, train on the same sample more than once, making it computationally infeasible to “overtrain” the model to eliminate hallucinations.
> Additionally, neural networks are smooth, and the learned diffusion model struggles to approximate the score function in regions with discontinuities or sharp changes. This limitation is inherent to the model architecture used. The proposed dynamic guidance uses an additional indicator variable (class) that switches between different classifier gradients. We hypothesize that this switching in the gradient added to the diffusion sampling step can help approximate these discontinuities better.

---

### Official Review · Reviewer_yFiB · 2025-10-29

**Soundness:** 2
**Presentation:** 3
**Contribution:** 2
**Rating:** 4
**Confidence:** 4

**Summary:**

The authors analyze hallucination in diffusion models and argue that it arises from interpolation between data modes. They hypothesize that the learned score function becomes too flat in these interpolation regions, causing the sampler to fall into traps and fail to escape. To mitigate this, they propose a selective guidance method that pushes samples toward the nearest class direction at each step, thereby steering them away from regions where hallucination typically occurs.

**Strengths:**

- **Easy-to-follow and intuitive approach:**
    The proposed method dynamically applies classifier guidance during unconditional generation, sharpening the distribution and easing generation away from mode-interpolated regions, thus reducing hallucination.

- **Well-designed toy examples:** The toy examples provides clear observations and insightful analysis, helping establish the problem motivation.

**Weaknesses:**

- **Limited experimental setting and weak scalability:**
    The method is only validated for unconditional generation. In class-conditional generation, if the nearest class differs from the desired target class, conflicts may arise, limiting applicability. Moreover, because the approach relies on classifier guidance, it is unlikely to extend to text-conditioned generation, where classes are not explicitly defined. Hallucination becomes even more problematic in more complex conditional scenarios, yet such cases are not explored.

- **Lack of metrics such as FID:**
    In ImageNet experiments, reduced recall relative to classifier guidance (CG) suggests limited class coverage, which the authors acknowledge as a limitation. For unconditional generation, it is crucial to verify whether all classes are sufficiently covered (assuming a balanced dataset). If class coverage deteriorates, this could undermine one of diffusion models’ strengths: broad mode coverage. It is unclear why FID is not reported, as it reflects both coverage and fidelity. Furthermore, if the method’s main advantage is improved mode-seeking, comparisons to GAN-like models that excel at this should be included.

- **Insufficient motivation behind selective sharpening:**
    Since the dynamic guidance pushes toward a specific class present in the training data (even if the class changes at each step), it is somewhat expected that this would reduce hallucination. However, the paper does not clearly explain why this method avoids interfering with interpolation unrelated to hallucination. While toy experiments show that latent dimensions unrelated to class are unaffected, a stronger theoretical or intuitive justification would be helpful.

**Questions:**

- If hallucination is also reflected in the classifier, the proposed method may not be able to address it. How do the authors view this issue?
- In Algorithm 1, dynamic guidance is applied between T1T_1T1 and T2T_2T2. These appear to be hyperparameters, but there is no explanation of how they are selected or any ablation.
- In L.726, there seems to be a typo. According to the description, the latent dimension for the position of the shape in the right figure should be 5, not 0.

---

> ### Author Response · Authors · 2025-11-22
> **Response to reviewer yFiB (1/3)**
>
> Thank you for your time and thoughtful suggestions, especially with regard to clarifying the theoretical justification for how our proposed Dynamic Guidance reduces interpolations that result in hallucinations while not affecting unrelated ones. Here we address each of your concerns:
>
> ### **1. Limited experimental setting and weak scalability**
>
> Knowing exact classes of hallucinations is not necessarily required - In **Single Shapes**, hallucinations directly correspond to interpolation between the chosen classes. However, in **Mixed Shapes**, this is not the case. We emphasize that DG does not require using a set of labels that correspond to actual classes or is identical to the set of labels used to control a generative model. The labels used for conditioning must correspond to human-interpretable semantics, such as object categories, attributes, or text, since they determine what the model is intended to generate. In contrast, the labels used for hallucination reduction do not need to carry such semantic meaning; they may correspond to abstract latent modes or auxiliary partitions of the data that help isolate hallucination-prone directions without mapping to interpretable concepts. We show that, for ImageNet, we can create pseudo-classes by clustering the training images using DINOv2. We train a classifier on those pseudo-classes. While the selected number of 5000 clusters does not directly correspond to the set of combinatorial subclasses of ImageNet labels, DG using pseudo-subclasses can still improve the model with regard to hallucinations. **We have added those results in the main text in Table 7 and created Section 5.2.2 to expand more on this discussion**.
>
> | Method                 | IS ↑  | Inception Prec ↑ | Inception Rec ↑ | CLIP Prec ↑ | CLIP Rec ↑ |
> |------------------------|-------|------------------|-----------------|-------------|-------------|
> | **Uncond. ADM**        | 34.24 | 0.53             | 0.61            | 0.60        | 0.26        |
> | **+ CG real classes**  | 83.19 | 0.69             | 0.55            | 0.74        | 0.27    |
> | **+ DG real classes**  | 88.49 | 0.77             | 0.52            | 0.77        | 0.26        |
> | **+ DG dino clusters** | 75.17 | 0.78             | 0.51            | 0.77        | 0.25        |
>
> ### **2. Lack of metrics such as FID**
>
> > It is unclear why FID is not reported...
>
> We would like to point the reviewer to **Appendix Section A.2.** where we discuss this topic. There, we computed Inception and CLIP FID scores, observing that in most settings, Dynamic Guidance outperforms Classifier Guidance. For high guidance scales ($\lambda = 10$), Dynamic Guidance is biased towards generating classes that are better matched with the initial noise, and thus ends up with non-uniform ImageNet classes in the generated samples.
> We point out that this heavily affects Inception FID since the ImageNet-1k-trained Inception model produces features heavily biased towards class-balanced ImageNet images. When using CLIP features to measure the FID, the difference is less significant.
> To fairly compare Dynamic Guidance to Classifier Guidance, we strictly enforce a balanced generated distribution of classes using rejection sampling. We classify each generated sample using the trained classifier and strictly enforce a uniform distribution of classes in the generated 10k image set. We emphasize that this class balancing is only performed using the existing classifier that is already used for guidance. When we compute the Inception FID with this balanced set, we find that Dynamic Guidance significantly outperforms Classifier Guidance.
>
> To conclude, we do agree with the reviewer that Dynamic Guidance can impact diversity, but this is not significant, as shown in Tables 3,8 and 9: our improvement in precision is much more pronounced than the reduction in recall, while CLIP recall is not reduced and generative coverage [1] (**which we have added to Appendix Section A.2. Table 9**) is increased.
>
> [1]: Naeem, Muhammad Ferjad, et al. "Reliable fidelity and diversity metrics for generative models." International conference on machine learning. PMLR, 2020.

---

> ### Author Response · Authors · 2025-11-22
> **Response to reviewer yFiB (2/3)**
>
> ### **3. Insufficient motivation behind selective sharpening**
>
> Dynamic Guidance relies on the gradient produced by the classifier. If this gradient corresponds to the classifier changing attributes related to the classification label, then we show that the altered score function only focuses on the dimensions where interpolation concerns misclassification (Single Shapes). **In Appendix Figure A.5 we add (b)** that shows that the gradients along hallucination-relevant directions are strong and informative, whereas gradients along irrelevant directions are noisy and close to zero.
>
> Additionally, diffusion models rely on neural networks that inherently represent smooth functions, which limits their ability to model regions where the true score exhibits sharp transitions or discontinuities, such as boundaries between semantic modes. These regions often correspond to hallucination-prone transitions in the data manifold. Dynamic Guidance introduces an external discrete signal (the class indicator), effectively switching between different classifier gradients across modes. We hypothesize that this discrete switching helps approximate sharp transitions in the score function that the diffusion model alone cannot capture due to architectural smoothness constraints. In this view, Dynamic Guidance can compensate for the challenges of representing those discontinuous boundaries by injecting discrete mode-specific directional information that sharpens the score selectively.
>
> ### **4. Hallucinations in classifier**
>
> > If hallucination is also reflected in the classifier, the proposed method may not be able to address it. How do the authors view this issue?
>
> The key to mitigating hallucinations is the classifier-produced gradient that we add at each step. By dynamically selecting the classifier target, we make sure that the gradient we obtain is noise-free and helpful towards pushing the sample away from unwanted regions. If we understood it correctly, the reviewer points out that if the classifier gradient fails to capture the 'hallucinated feature', then we cannot help the generation with avoiding it. For instance, if a hallucinated image is classified with 100\% accuracy, then the classifier gradient would be zero, which translates to no changes made to the sampling. We agree that this is a limitation, but would like to argue that it is unlikely we will end up in this situation, even with a weakly-trained classifier. We would like to point the reviewer to our answer to his first question. Our experiment using pseudo-classes created by clustering DINOv2 embeddings uses a classifier that is weaker than the one trained on real ImageNet classes, but is still effective in mitigating hallucinations.
>
> ### **5. Why is guidance applied in an interval [T1,T2]**
>
> We identify that, unlike Classifier Guidance, our proposed Dynamic Guidance (DG) can be performed just for a subset of timesteps, which we denote in our algorithm as [$T_1,T_2$]. Classifier Guidance attempts to impose a \textbf{strong constraint} on the generation process: the final sample should belong to the chosen class. On the other hand, Dynamic Guidance is more similar to Classifer-Free Guidance (CFG) in that it attempts to guide a strong signal (conditional model in the case of CFG, unconditional model in the case of DG) with a weak guidance signal (unconditional model in the case of CFG, dynamic gradient of a classifier in the case of DG) to improve generation.
>
> Inspired by recent work on CFG [1,2], we show that applying DG only for some intermediate timesteps [$T_1,T_2$] improves performance. To select $T_1$ and $T_2$ for each experiment, we chose the generation timestep at which the image begins to form ($T_1$), and the timestep where samples appear to have already converged to an image that cannot be modified further ($T_2$). We verify our choice of $[T_1,T_2]$ in Table 6, where for our ImageNet experiments the selected $T_1=800$ and $T_2=400$ achieve the best results. In Figures A.6 and A.7 in the Appendix, we visualize the generation process to show how our choice of $T_1$ and $T_2$ is motivated by the image formation process.
>
> | Method                         | Inception Score ↑ | Precision ↑ | Recall ↑ |
> |--------------------------------|------------------:|------------:|----------:|
> | **Uncond. ADM**                | 34.24             | 0.53        | 0.61      |
> | **+ Dynamic Guidance 1000–0**  | 48.20             | 0.73        | 0.33      |
> | **+ Dynamic Guidance 600–200** | 61.07             | 0.64        | **0.62**     |
> | **+ Dynamic Guidance 800–400** | **88.49**         | **0.77**    | 0.52  |
>
> We add this table and explanation in Section 5.2.1 of the main text.
>
> [1]: Kynkäänniemi, Tuomas, et al. "Applying guidance in a limited interval improves sample and distribution quality in diffusion models." NeurIPS (2024).
> [2]: Wang, Xi, et al. "Analysis of classifier-free guidance weight schedulers." TMLR (2024).

---

> ### Author Response · Authors · 2025-11-22
> **Response to reviewer yFiB (3/3)**
>
> ### **6. Typos**
>
> > In L.726, there seems to be a typo. According to the description, the latent dimension for the position of the shape in the right figure should be 5, not 0.
>
> We thank the reviewer for finding this typo; it should indeed be 5. We have corrected this in the revised version.

---

> > ### Comment · Reviewer_yFiB · 2025-11-25
> >
> > Thank you for your detailed response. I sincerely appreciate the effort put into addressing my comments and the hard work involved in revising the manuscript.
> >
> > While the rebuttal has resolved some of my concerns, a one point remains.
> >
> > I think the demonstration of utilizing non-semantic labels, such as DINO clustering classes, to be quite impressive; this effectively alleviates my initial concern regarding the strict necessity of semantic labels.
> >
> > However, my original concern still remains beyond the unconditional setting primarily discussed in your response. specifically regarding 'class-conditional generation,' my concern is not only about low class coverage or induced bias but also about the fundamental issue of whether the model can correctly generate the "intended class." If the dynamic guidance steers the sample towards a different class than the condition, it poses a problem of faithfulness.
> >
> > Furthermore, while DINO clustering label can be meaningful direction, searching within such a constrained space still seems to lack sufficient scalability compared to the vast, open-ended knowledge representation required for modern Text-to-Image models.

---

> > > ### Author Response · Authors · 2025-11-26
> > > **Response to Reviewer yFiB regarding class-conditional generation**
> > >
> > > Thank you for your response. We are glad that most of your concerns have been addressed.
> > >
> > > Regarding conditional generation with Dynamic Guidance, of course, if we use the same labels for conditioning and guidance, the two will end up agreeing, diminishing the effect of dynamic guidance. However, **we add Figure A.19** in the Appendix to show that when we have different sets (ImageNet class-conditioned model, Dynamic Guidance with DINOv2 generated cluster labels), the two can act orthogonally, with class-conditioning guiding the sample towards specific class-related attributes, and dynamic guidance helping avoid bad generations within the class.
> > >
> > > We leave the in-depth investigation of how to effectively choose labels for Dynamic Guidance in conditional settings and hyper-parameter tuning to future work.
> > >
> > > > Furthermore, while DINO clustering label can be meaningful direction, searching within such a constrained space still seems to lack sufficient scalability compared to the vast, open-ended knowledge representation required for modern Text-to-Image models.
> > >
> > > We show (Figure A.19) that the conditioned model and Dynamic Guidance can work together effectively even when using disjoint sets of labels; this means that having to pick the labels for Dynamic Guidance does not necessarily constrain the Diffusion Model to using those labels; it can be conditioned on a different set or even text.

---

### Official Review · Reviewer_oYdW · 2025-11-01

**Soundness:** 2
**Presentation:** 3
**Contribution:** 3
**Rating:** 4
**Confidence:** 4

**Summary:**

The paper introduces a new sampling approach for diffusion models that they call dynamic guidance. This is introduced to alleviate the mode interpolation based hallucination issues identified in Aithal et. al. 2024. The key idea is that dynamic guidance adjusts classifier guidance during sampling rather than using a fixed condition. This is done by re-aligning the guidance condition to the most likely mode at each denoising step. The authors find that the new method reduces hallucinations close to 70%, and improves precision on Imagnet.

**Strengths:**

1. The score function sharpening approach is well founded both mathematically and emprically (as see in Figure 3)
2. The dynamic scoring approach achieves better fidelity for both DDPM and DDIM when steps are small.
3. The evaluation is diverse across a set of toy + hands + imagenet tasks.

**Weaknesses:**

## Diversity Discussion
1. My biggest concern with this work is that there is no explicit discussion on diversity (or model) collapse. This is almost central to the work. Diversity and Interpolation are fundamentally at odds, and an investigation into the reduction of interpolation is incomplete without measuring its impact on diversity.
2. Of note, the paper briefly reports precision and recall metrics on ImageNet (Table 3) and shows some qualitative examples. It can be seen that the recall drops (e.g., 0.61 → 0.52), and Appendix Figures A.15–A.16 reveal that Dynamic Guidance leads to over-representation of certain ImageNet classes.
3. This indicates that while hallucinations are reduced, the model’s generative support becomes narrower, favoring a subset of modes where the classifier is more confident.
4. For hallucination-mitigation research, measuring this trade-off is essential, as overly “safe” sampling may degrade the creative or exploratory capacity of generative models.


## Need for a Classifier
I am concerned if we will always be able to know the classes of hallucination that the classifier could use for hallucination-prone regions. I am not sure how this approach will scale beyond toy tasks. Could you throw more light on its expansion to general purpose models?

**Questions:**

Can you include explicit diversity metrics such as LPIPS variance, embedding-space entropy, coverage/recall FID curves to quantify how Dynamic Guidance affects the distributional breadth of outputs.

---

> ### Author Response · Authors · 2025-11-22
> **Response to reviewer oYdW**
>
> Thank you for your time and thoughtful suggestions. We are glad you found our approach mathematically and empirically grounded. Here we address each of your concerns:
>
> ### **1. Diversity Discussion**
>
> We agree with the reviewer that since our method discusses interpolations and their reduction (or mitigation), discussing the impact on diversity is central to it. This is the aim of the existing limitations section, where we address potential over- and under-representation of classes. We rewrote this section, making it a more explicit discussion on the impact of our method on diversity of generations. Additionally, we want to highlight the role of choosing classes on the diversity-hallucination tradeoff. When one can pick classes whose direct interpolations will lead to hallucinations, then we have shown that there is **no tradeoff**, as in the example of the Single Shapes dataset. There, our guidance sharpens the score function in the latent directions corresponding to hallucinations but does not affect it elsewhere, preserving diversity in generations. We then show that even in the harder case where we cannot pick guidance classes so that interpolation between them directly aligns with what we have defined as a hallucination, we can still be effective in reducing hallucinations, albeit at the cost of also hurting diversity.
>
> > Can you include explicit diversity metrics such as LPIPS variance, embedding-space entropy, coverage/recall FID curves to quantify how Dynamic Guidance affects the distributional breadth of outputs.
>
> We thank the reviewer for this suggestion. Regarding LPIPS variance, we were unable to find any mentions of it being used to measure diversity in generative models, and kindly ask for clarifications as to what the reviewer would like to see.
> We provide a table for calculated density and coverage [1] based on CLIP. We see that Dynamic Guidance improves the density of the generations from 0.70 to 0.97 while not sacrificing coverage (0.66 vs 0.67). **We include this table in Section A.2. in the Appendix.**
>
> | Method              | Density ↑ | Coverage ↑ |
> |---------------------|-----------|-------------|
> | **Uncond.**         | 0.7003    | 0.6662      |
> | **+CG (λ=1)**       | 0.5366    | 0.4473      |
> | **+DG (λ=1)**       | 0.7023    | 0.5921      |
> | **+CG (λ=10)**      | 0.9058    | 0.7516      |
> | **+DG (λ=10)**      | 0.9724    | 0.6761      |
> | **+CG (λ=20)**      | 0.7999    | 0.6943      |
> | **+DG (λ=20)**      | 0.9285    | 0.6385      |
>
>
> [1]: Naeem, Muhammad Ferjad, et al. "Reliable fidelity and diversity metrics for generative models." International conference on machine learning. PMLR, 2020.
>
> ### **2. Need for a classifier - Expansion to general purpose models**
>
> Knowing exact classes of hallucinations is not necessarily required - In **Single Shapes**, hallucinations directly correspond to interpolation between the chosen classes. However, in **Mixed Shapes**, this is not the case. We emphasize that DG does not require using a set of labels that correspond to actual classes or is identical to the set of labels used to control a generative model. The labels used for conditioning must correspond to human-interpretable semantics, such as object categories, attributes, or text, since they determine what the model is intended to generate. In contrast, the labels used for hallucination reduction do not need to carry such semantic meaning; they may correspond to abstract latent modes or auxiliary partitions of the data that help isolate hallucination-prone directions without mapping to interpretable concepts. We show that, for ImageNet, we can create pseudo-classes by clustering the training images using DINOv2. We train a classifier on those pseudo-classes. While the selected number of 5000 clusters does not directly correspond to the set of combinatorial subclasses of ImageNet labels, DG using pseudo-subclasses can still improve the model with regard to hallucinations. **We have added those results in the main text in Table 7 and created Section 5.2.2 to expand more on this discussion**.
>
> | Method                 | IS ↑  | Inception Prec ↑ | Inception Rec ↑ | CLIP Prec ↑ | CLIP Rec ↑ |
> |------------------------|-------|------------------|-----------------|-------------|-------------|
> | **Uncond. ADM**        | 34.24 | 0.53             | 0.61            | 0.60        | 0.26        |
> | **+ CG real classes**  | 83.19 | 0.69             | 0.55            | 0.74        | 0.27    |
> | **+ DG real classes**  | 88.49 | 0.77             | 0.52            | 0.77        | 0.26        |
> | **+ DG dino clusters** | 75.17 | 0.78             | 0.51            | 0.77        | 0.25        |

---

### Official Review · Reviewer_iNsy · 2025-11-02

**Soundness:** 4
**Presentation:** 4
**Contribution:** 3
**Rating:** 6
**Confidence:** 4

**Summary:**

This paper aims to reduce hallucinations in diffusion generations. The authors identify the root cause as excessive smoothing of the learned score function between distinct modes of the data distribution, which leads to mode interpolation and unrealistic generations. To mitigate this, the authors propose Dynamic Guidance that continually re-estimates the most likely class based on the current noisy sample and incurs a classifier-based guidance for that class.
The paper evaluates the method on several settings: toy 2D Gaussian mixtures, synthetic shape datasets, human hand datasets, and large-scale ImageNet. Results show consistent hallucination reduction across different datasets.

**Strengths:**

1. The paper is very clearly-written and easy to follow. The dynamic guidance is an easy-to-implement and elegant technique in practice. The paper presented good visualizations (e.g. Figure 1 & 3) to exhibit the sharpening effect of dynamic guidance.

2. The paper studies an important topic for diffusion models and presents strong empirical evidence for the effectiveness of the guidance.

3. The paper presented certain limitations which is also the reviewer's concern.

**Weaknesses:**

1. The method of the paper is limited to solve hallucinating in certain realistic scenarios when there is too much classes (e.g. combinatorial classes https://openreview.net/forum?id=SKW10XJlAI) or continuous gestures of the hands that may not fall into discrete classes. The classifier-based methods are also training-expensive in the sense that we need a well-trained classifier at every noise level for good model performance. These limit the applicability of the technique in practice.

2. In the experiments, the paper exhibits that while classifier-based guidance is ineffective to reduce hallucination in some cases, dynamic guidance is effective. Due to the similarity between the two types of guidance, the paper should delve deeper into such a distinction, which is its key novelty and contribution. However not much observations are made in the paper like how "dynamic" is the classifying process and why dynamicity is crucial to reduce hallucination.

3. The paper does not contrast its technique with many existing baselines (e.g. https://arxiv.org/pdf/2402.08680) and sampling techniques. It also does not account for solving the limitations using methods like classifier calibrations.

**Questions:**

1. Why is the guidance only applied in an interval $[T_1,T_2]$ in the generation process in stead of the whole process? How are $T_1, T_2$ selected?

2. Why cannot classifier guidance eliminate mode interpolations?

---

> ### Author Response · Authors · 2025-11-22
> **Response to reviewer iNsy (1/2)**
>
> Thank you for your time and thoughtful suggestions, especially with regard to clarifying the limitations of Classifier Guidance compared to our proposed Dynamic Guidance. Here we address each of your concerns:
> ### **1. Limitation regarding number of classes, or continuous classes**
>
> We emphasize that DG does not require using a set of labels that correspond to actual classes or is identical to the set of labels used to control a generative model. The labels used for conditioning must correspond to human-interpretable semantics, such as object categories, attributes, or text, since they determine what the model is intended to generate. In contrast, the labels used for hallucination reduction do not need to carry such semantic meaning; they may correspond to abstract latent modes or auxiliary partitions of the data that help isolate hallucination-prone directions without mapping to interpretable concepts. This means that for the situation of an extreme amount of combinatorial classes, we don't need to define and use all of them. We show that, for ImageNet, we can create pseudo-classes by clustering the training images using DINOv2. We train a classifier on those pseudo-classes. While the selected number of 5000 clusters does not directly correspond to the set of combinatorial subclasses of ImageNet labels, DG using pseudo-subclasses can still improve the model with regard to hallucinations. We have added those results in the main text in Table 7 and created Section 5.2.2 to expand more on this discussion.
>
> | Method                 | IS ↑  | Inception Prec ↑ | Inception Rec ↑ | CLIP Prec ↑ | CLIP Rec ↑ |
> |------------------------|-------|------------------|-----------------|-------------|-------------|
> | **Uncond. ADM**        | 34.24 | 0.53             | 0.61            | 0.60        | 0.26        |
> | **+ CG real classes**  | 83.19 | 0.69             | 0.55            | 0.74        | 0.27    |
> | **+ DG real classes**  | 88.49 | 0.77             | 0.52            | 0.77        | 0.26        |
> | **+ DG dino clusters** | 75.17 | 0.78             | 0.51            | 0.77        | 0.25        |
>
>
> ### **2. Why does Classifier Guidance fail?**
>
> While classifier-based guidance can, in principle, direct samples toward avoiding intermediate regions between modes, it requires an accurate signal from the classifier to achieve this. The initial noise sample can be far from the chosen target mode, making the classifier gradients noisy with high variance. As a result, the guidance signal is inaccurate and not effective in guiding the sample away from the regions between modes. Additionally, when the sample is far from the intended mode, the guidance must make large steps to correct the predicted class label. Those large guidance steps are based on noisy, inaccurate gradients, amplifying the error and overshooting or drifting into intermediate regions. In contrast, dynamic guidance adaptively selects the closest mode at each sampling step, ensuring that guidance is always applied locally, where classifier gradients are informative and the required updates smaller.
>
> To illustrate this, we pick the initial noise sample that gives us the class 'bison' when sampling with Dynamic Guidance (**Figure A.6 in the Appendix**). When sampling with Classifier Guidance and choose the class ''llama'', the diffusion model, in the early timesteps, has created a large circular blob in the center of the image. To correct this (since it does not resemble a llama), the applied guidance attempts to change the image by changing the shape of the generated object and adding thin legs. However, since the change required is too large, the guidance ends up being inaccurate, and the model produces a hallucinated sample. In practice, when the sample ends up close to multiple modes, then the selected mode changes a lot during the sampling process (**Figure A.7 in the Appendix**).
>
> ### **3. Referenced baseline - Classifier calibration**
>
> The reference baseline tackles a **different problem**, object hallucination in the generated **text** of autoregressive VLMs. It uses guidance to alter the predicted probabilities of the discrete tokens in text space. That method is not directly applicable to score-based models, and does not refer to an image-generation task. We add this as a reference to the recent hallucination mitigation literature, but are still unsure as to how to compare it with our method, which focuses on score-based (diffusion) models and image generation.
>
> Regarding classifier calibration, we would like to ask the reviewer whether they had a specific algorithm or method in mind. In general, the calibration of the classifier we use for guidance does not affect our algorithm, since we always use the top-predicted class for guidance. A badly calibrated classifier would still provide useful gradients for its top-predicted class.

---

> ### Author Response · Authors · 2025-11-22
> **Response to reviewer iNsy (2/2)**
>
> ### **4. Why is guidance applied in an interval [T1,T2]**
>
> We identify that, unlike Classifier Guidance, our proposed Dynamic Guidance (DG) can be performed just for a subset of timesteps, which we denote in our algorithm as [$T_1,T_2$]. Classifier Guidance attempts to impose a \textbf{strong constraint} on the generation process: the final sample should belong to the chosen class. On the other hand, Dynamic Guidance is more similar to Classifer-Free Guidance (CFG) in that it attempts to guide a strong signal (conditional model in the case of CFG, unconditional model in the case of DG) with a weak guidance signal (unconditional model in the case of CFG, dynamic gradient of a classifier in the case of DG) to improve generation.
>
> Inspired by recent work on CFG [1,2], we show that applying DG only for some intermediate timesteps [$T_1,T_2$] improves performance. To select $T_1$ and $T_2$ for each experiment, we chose the generation timestep at which the image begins to form ($T_1$), and the timestep where samples appear to have already converged to an image that cannot be modified further ($T_2$). We verify our choice of $[T_1,T_2]$ in Table 6, where for our ImageNet experiments the selected $T_1=800$ and $T_2=400$ achieve the best results. In Figures A.6 and A.7 in the Appendix, we visualize the generation process to show how our choice of $T_1$ and $T_2$ is motivated by the image formation process.
>
> | Method                         | Inception Score ↑ | Precision ↑ | Recall ↑ |
> |--------------------------------|------------------:|------------:|----------:|
> | **Uncond. ADM**                | 34.24             | 0.53        | 0.61      |
> | **+ Dynamic Guidance 1000–0**  | 48.20             | 0.73        | 0.33      |
> | **+ Dynamic Guidance 600–200** | 61.07             | 0.64        | **0.62**     |
> | **+ Dynamic Guidance 800–400** | **88.49**         | **0.77**    | 0.52  |
>
> We add this table and explanation in Section 5.2.1 of the main text.
>
> [1]: Kynkäänniemi, Tuomas, et al. "Applying guidance in a limited interval improves sample and distribution quality in diffusion models." Advances in Neural Information Processing Systems 37 (2024): 122458-122483.
> [2]: Wang, Xi, et al. "Analysis of classifier-free guidance weight schedulers." arXiv preprint arXiv:2404.13040 (2024).

---

> > ### Comment · Reviewer_iNsy · 2025-11-28
> >
> > I would appreciate the authors for the reply and would like to increase my rating of the paper given that the illustrations in the reply are incorporated into the final version of the paper. (However I don't know how to edit the above review.) Overall I think the paper is presenting interesting findings that should make it accepted to the conference.

---

> > > ### Author Response · Authors · 2025-12-02
> > > **Response to Reviewer iNsy**
> > >
> > > Thank you for being willing to increase your score (even if not possible given the situation), we are glad all your concerns have been addressed. Thank you for your constructive feedback; all changes are also included in the manuscript in red and will also be included in the final version of the paper.

---

### Author Response · Authors · 2025-11-22
**Response to all reviewers**

We thank all the reviewers for their time and effort. **We have updated the manuscript to address the reviewers' comments; edits are marked in red.**

---

### Author Response · Authors · 2025-12-02
**Summary of discussion phase**

We would like to thank the reviewers and Area Chair for the effort and time commited to evaluating and reviewing our manuscript. We have addressed each of the reviewers concerns and updated the manuscript accordingly with further analysis and experimental evaluation.

**In the hopes of aiding the process, we provide a concise summary of the author-reviewer discussion.**
* * *

## Reviewer iNsy

| Major Questions    | Our response and actions |
|----------|--------|
| Number of classes | We emphasize that DG does not require using the same set of classes as those that would be used for conditioning or that the classes used be human interpretable. We add an experiment with DG using a classifier trained on classes created by clustering DINOv2 embeddings (**Section 5.2.2, Table 7**). |
| Why does Classifier Guidance fail?  | We provide further explanation on this and add qualitative analysis to the Appendix (**Figures A.6, A.7**).|
| Why apply guidance at an interval \[T1,T2\]?  | We explain our motivation on applying guidance at an interval and add an ablation study for our choice of T1,T2 (**Section 5.2.1, Figures A.6, A.7**).|

Reviewer iNsy was satisfied with our clarifications and expressed intent to raise his score.

## Reviewer oYdW

| Major Questions    | Our response and actions |
|----------|--------|
| Concern regarding diversity in generation | We edited our existing limitations section to more explicitly discuss the impact on diversity in **Section 6**, explaining the cases where there is **no tradeoff**, and the cases where diversity might be impacted. We also provide additional experiments on ImageNet, showing that our method does not impact diversity significantly (**Section A.2**).|
| Need for a classifier/explicit classes  | We emphasize that DG does not require knowing the exact classes of hallucinations. We add an experiment with DG using a classifier trained on classes created by clustering DINOv2 embeddings and a section on the main paper that discusses this further(**Section 5.2.2, Table 7**).|

## Reviewer yFiB

| Major Questions    | Our response and actions |
|----------|--------|
| Scalability/necessity of strict semantic labels | We emphasize that DG does not require knowing the exact classes of hallucinations. We add an experiment with DG using a classifier trained on classes created by clustering DINOv2 embeddings and a section on the main paper that discusses this further(**Section 5.2.2, Table 7**). |
| Why apply guidance at an interval \[T1,T2\]?  | We explain our motivation on applying guidance at an interval and add an ablation study for our choice of T1,T2 (**Section 5.2.1, Figures A.6, A.7**).|

Reviewer yFiB was satisfied with our clarifications and had one remaining concern regarding conditional generation. We addressed this in another response:

| Major Questions    | Our response and actions |
|----------|--------|
| Conditional generation | We explain that if the same set of labels are used for conditioning and guidance, then of course they will end up agreeing, diminishing the effect of DG. We add **Figure A.19** to show what happens that when we use distinct sets (ImageNet classes vs DINO clusters) the two can act orthogonally and DG can improve generations even in the class-conditional case.|


## Reviewer Qih7

| Major Questions    | Our response and actions |
|----------|--------|
| Connection to Classifier-Free Guidance | We emphasize that our work is orthogonal to CFG; the proposed Dynamic Guidance (DG) does not require using the same set of classes. We provide additional experiments with DG using classes created by clustering DINOv2 embeddings on an unconditional or ImageNet-class-conditional model (**Section 5.2.2, Table 7, Figure A.19**) to show that DG and conditioning can be used with distinct sets of classes to improves generations. This means that one can potentially use CFG and DG together with disjoint sets of classes.|
| Theory behind score sharpening  | We additionally clarify how DG selectively targets latent dimensions associated with hallucinations while leaving unrelated dimensions unaffected. To illustrate this behavior, we include further examples (**Figure A.5b**) showing how the gradients produced by DG differ in hallucination-related directions compared to unrelated ones. |


Reviewer Qih7 wanted to see our method applied on text-to-image models like Stable Diffusion. This would require a significant effort to decide on the proper hyperparameters and exact way of applying to a text-conditioned model setting. In place of that we provided a class-conditional example (**Figure A.19**, discussion with reviewer yFiB). The reviewer was partially satisfied and expressed willingness to raise his score.
* * *
All changes described have been incorporated in the manuscript in red. We hope this summary is helpful in the meta-review process.

---

### Meta-Review · Area_Chair_rrtS · 2025-12-13

**Summary:**

Thanks to the authors for providing a very good summary of the reviewers' concerns and how they addressed them.

The reviewer's assessment of this paper is rather broad. While the reviewer iNsy indicated that they raised the score (probably to 8), the other three reviewers remained with 4, 4, and 6. Reviewer Qih7 and yFiB mentioned they would like to see the application of DG to T2I cases, which I would also hope to see, to judge whether the proposed method is really practical in more realistic settings than class-conditional or unconditional cases.  Hence, I recommend rejecting this paper.

**Reviewer Concerns:**

* Diversity: The authors showed that DG does not harm diversity of generated samples in their additional experiment.

* Scalability/necessity of strict semantic labels or perfect classifiers: The authors performed an additional experiment to discuss this point.

**Reviewer Scores:**

* Reviewer Qih7 already mentioned they would bump it up to 7 if there was such an option. But the reviewer is hesitant to give 8, as they require T2I experiments to be fully convinced.

* Reviewer yFiB: The reviewer expressed partial satisfaction in their remark. However, I sense that this reviewer is also quite hesitant to put "full accept," as they require T2I experiments to be fully convinced.

---

### Decision · Program_Chairs · 2026-01-26

Reject